# Enhancing Certified Robustness via Block Reflector Orthogonal Layers

## Abstract

Lipschitz neural networks are well-known for providing certified robustness in deep learning. In this paper, we present a novel efficient Block Reflector Orthogonal layer that enables the construction of simple yet effective Lipschitz neural networks. In addition, by theoretically analyzing the nature of Lipschitz neural networks, we introduce a new loss function that employs an annealing mechanism to improve margin for most data points. This enables Lipschitz models to provide better certified robustness. By employing our BRO layer and loss function, we design BRONet, which provides state-of-the-art certified robustness. Extensive experiments and empirical analysis on CIFAR-10, CIFAR-100, and Tiny-ImageNet validate that our method outperforms existing baselines. [1]

## 1 Introduction

Although deep learning has been widely adopted in various fields (Wang et al., 2022; Brown et al., 2020), it is shown to be vulnerable to adversarial attacks (Szegedy et al., 2013). This kind of attack crafts an imperceptible perturbation on images (Goodfellow et al., 2014) or voices (Carlini & Wagner, 2018) to make AI systems make incorrect predictions. In light of this, many adversarial defense methods have been proposed to improve the robustness, which can be categorized into empirical defenses and theoretical defenses. Common empirical defenses include adversarial training (Madry et al., 2018; Shafahi et al., 2019; Wang et al., 2023) and preprocessing-based methods (Samangouei et al., 2018; Das et al., 2018; Lee & Kim, 2023). Though effective, the empirical defenses cannot provide any robustness guarantees. Thus, the defenses may be ineffective when encountering sophisticated attackers. Unlike empirical defenses, theoretical defenses offer quantitative and provable guarantees of robustness, ensuring no adversarial examples within a specific $\ell_p$-norm ball with a radius $\varepsilon$ around the prediction point.

Theoretical defenses against adversarial attacks are broadly categorized into *probabilistic* and *deterministic* (Li et al., 2023) methods. Randomized smoothing (Cohen et al., 2019; Lecuyer et al., 2019; Yang et al., 2020) is a prominent probabilistic approach, known for its scalability in providing certified robustness. However, its reliance on extensive sampling substantially increases computational overhead during inference, limiting its practical deployment. Furthermore, the certification provided is probabilistic in nature.

Conversely, deterministic methods, exemplified by interval bound propagation (Ehlers, 2017; Gowal et al., 2018; Mueller et al., 2022; Shi et al., 2022) and CROWN (Wang et al., 2021; Zhang et al., 2022), efficiently provide deterministic certification. These methods aim to approximate the lower bound of worst-case robust accuracy to ensure deterministic robustness guarantees. Among various deterministic methods, neural networks with Lipschitz constraints are able to compute the lower bound of worst-case robust accuracy with a single forward pass, making them the most time-efficient at inference time. They are known as Lipschitz neural networks.

Lipschitz neural networks are designed to ensure that the entire network remains Lipschitz-bounded. This constraint limits the sensitivity of the outputs to input perturbations, thus

---

[1]The code will be made available upon acceptance. A version has been provided for reviewers.

providing certifiable robustness by controlling changes in the logits. A promising approach to constructing Lipschitz neural networks focuses on designing orthogonal layers, which inherently satisfy the the 1-Lipschitz constraint. Furthermore, these layers help mitigate the issue of vanishing gradient norms due to their norm-preserving properties.

In this work, we introduce the **Block Reflector Orthogonal (BRO)** layer, which outperforms existing methods in terms of computational efficiency as well as robust and clean accuracy. We utilize our BRO layer to develop various Lipschitz neural networks, thereby demonstrating its practical utility across various architectures. Additionally, we develop a new Lipschitz neural network **BRONet**, which shows promising results.

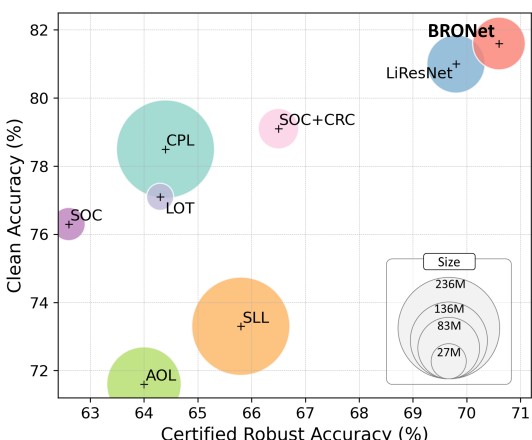

Figure 1: Visualization of model performance. The circle size denotes model size.

Moreover, we delve into Lipschitz neural networks, analyzing their inherent limited capability. Building on this analysis, we introduce a novel loss function, the *Logit Annealing* loss, which is empirically shown to be highly effective for training Lipschitz neural networks. The certification results of the proposed method outperform state-of-the-art methods with reasonable number of parameters, as Figure 1 shows.

Our contributions are summarized as follows:

- We propose a novel BRO method to construct orthogonal layers using low-rank parameterization. It is both time and memory efficient, while also being stable during training by eliminating the need for iterative approximation algorithms.

- We unlock the potential of applying orthogonal layers to more advanced architectures, enhancing certified robustness while reducing resource requirements.

- We construct various Lipschitz networks using BRO method, including newly designed BRONet, which achieves state-of-the-art certified robustness without adversarial training.

- Based on our theoretical analysis, we develop a novel loss function, the Logit Annealing loss, which is effective for training Lipschitz neural networks via an annealing mechanism.

- Through extensive experiments, we demonstrate the effectiveness of our proposed method on the CIFAR-10, CIFAR-100, and Tiny-ImageNet datasets.

## 2 PRELIMINARIES

### 2.1 CERTIFIED ROBUSTNESS WITH LIPSCHITZ NEURAL NETWORKS

Consider a function $f : \mathbb{R}^m \to \mathbb{R}^n$. The function is said to exhibit $L$-Lipschitz continuity under the $\ell_2$-norm if there exists a non-negative constant $L$ such that:

$$L = \text{Lip}(f) = \sup_{x_1, x_2 \in \mathbb{R}^m} \frac{\|f(x_1) - f(x_2)\|}{\|x_1 - x_2\|}, \tag{1}$$

where $\|\cdot\|$ represents the $\ell_2$ norm. This relationship indicates that any variation in the network's output is limited to at most $L$ times the variation in its input, effectively characterizing the network's stability and sensitivity to input changes. Specifically, under the $\ell_2$-norm, the Lipschitz constant is equivalent to the spectral norm of the function's Jacobian matrix.

Assuming $f(x)$ is the output logits of a neural network, and $t$ denotes the target label. We say $f(x)$ is certifiably robust with a certified radius $\varepsilon$ if $\arg\max_i f(x + \delta)_i = t$ for all perturbations $\{\delta : \|\delta\| \leq \varepsilon\}$. Determining the certified radii is crucial for certifiable robustness and presents a significant challenge. However, in $L$-Lipschitz neural networks,

$\varepsilon$ can be easily calculated using $\varepsilon = \max(0, \ \mathcal{M}_f(x)/\sqrt{2}L)$, where $\mathcal{M}_f(x)$ denotes the logit difference between the ground-truth class and the runner-up class in the network output. That is, $\mathcal{M}_f(x) = f(x)_t - \max_{k \neq t} f(x)_k$ (Tsuzuku et al., 2018; Li et al., 2019).

## 2.2 Lipschitz Constant Control & Orthogonality in Neural Networks

Obtaining the exact Lipschitz constant for general neural networks is known to be an NP-hard problem (Virmaux & Scaman, 2018). However, there are efficient methods available for computing it on a layer-by-layer basis. Once the Lipschitz constant for each layer is determined, the Lipschitz composition property allows for the calculation of the overall Lipschitz constant for the entire neural network. The Lipschitz composition property states that given two functions $f$ and $g$ with Lipschitz constants $L_f$ and $L_g$, their composition $h = g \circ f$ is also Lipschitz with a constant $L_h \leq L_g \cdot L_f$. We can use this property to obtain the Lipschitz constant of a complex neural network $f$:

$$f = \phi_l \circ \phi_{l-1} \circ \ldots \circ \phi_1, \quad \text{Lip}(f) \leq \prod_{i=1}^{l} \text{Lip}(\phi_i). \tag{2}$$

Thus, if the Lipschitz constant of each layer is properly regulated, robust certification can be provided. A key relevant property is orthogonality, characterized by the isometry property $\|Wx\| = \|x\|$ for a given operator $W$. Encouraging orthogonality is crucial for controlling the Lipschitz constant while preserving model expressiveness (Anil et al., 2019), as it helps mitigate the vanishing gradient problem and ensures a tight Lipschitz bound for the composition of layers in Equation 2.

## 3 Related Work

**Orthogonal Layers** Orthogonality in neural networks is crucial for various applications, including certified robustness via Lipschitz-based methods, GAN stability (Müller et al., 2019), and training very deep networks with inherent gradient preservation. While some approaches implicitly encourage orthogonality through regularization or initialization (Qi et al., 2020; Xiao et al., 2018), explicit methods for constructing orthogonal layers have garnered significant attention, as evidenced by several focused studies in this area. Li et al. (2019) proposed *Block Convolution Orthogonal Parameterization (BCOP)*, which utilizes an iterative algorithm for orthogonalizing the linear transformation within a convolution. Trockman & Kolter (2020) introduced a method employing the *Cayley transformation* $W = (I - V)(I + V)^{-1}$, where $V$ is a skew-symmetric matrix. Similarly, Singla & Feizi (2021b) developed the *Skew-Orthogonal Convolution (SOC)*, employing an exponential convolution mechanism for feature extraction. Additionally, Xu et al. (2022) proposed the *Layer-wise Orthogonal training (LOT)*, an analytical solution to the orthogonal Procrustes problem (Schönemann, 1966), formulated as $W = (VV^T)^{-1/2}V$. This approach requires the Newton method to approximate the internal matrix square root. Yu et al. (2021) proposed the *Explicitly Constructed Orthogonal Convolution (ECO)* to enforce all singular values of the convolution layer's Jacobian to be one. Notably, SOC and LOT achieve state-of-the-art certified robustness for orthogonal layers. Most matrix re-parameterization-based methods can be easily applied for dense layers, such as Cayley, SOC, and LOT. One recently proposed orthogonalization method for dense layers is *Cholesky* (Hu et al., 2024), which explicitly performs QR decomposition on the weight matrix via Cholesky decomposition.

**Other 1-Lipschitz Layers** A relaxation of isometry constraints, namely, $\|Wx\| \leq \|x\|$, facilitates the development of extensions to orthogonal layers, which are 1-Lipschitz layers. Prach & Lampert (2022) introduced the *Almost Orthogonal Layer (AOL)*, which is a rescaling-based parameterization method. Meanwhile, Meunier et al. (2022) proposed the *Convex Potential Layer (CPL)*, leveraging convex potential flows to construct 1-Lipschitz layers. Building on CPL, Araujo et al. (2023) presented *SDP-based Lipschitz Layers (SLL)*, incorporating AOL constraints for norm control. Most recently, Wang & Manchester (2023) introduced the Sandwich layer, a direct parameterization that analytically satisfies the SDP conditions outlined by Fazlyab et al. (2019).

**Lipschitz Regularization** While the aforementioned methods control Lipschitz constant by formulating constrained layers with guaranteed Lipschitz bound, Lipschitz regularization methods estimate the layer-wise Lipschitz constant via power iteration (Farnia et al., 2019) and apply regularization to control it. Leino et al. (2021) employed a Lipschitz regularization term to maximize the margin between the ground truth and runner-up class in the loss function. Hu et al. (2023; 2024) further proposed a new Lipschitz regularization method *Efficiently Margin Maximization (EMMA)*, which dynamically adjust all the non-ground-truth logits before calculating the cross-entropy loss.

## 4 BRO: Block Reflector Orthogonal Layer

In this section, we introduce the BRO layer, designed to provide certified robustness via low-rank orthogonal parameterization. First, we detail the fundamental properties of our method. Next, we leverage the 2D-convolution theorem to develop the BRO orthogonal convolutional layer. Finally, we conduct a comparative analysis of our BRO with existing state-of-the-art orthogonal layers.

### 4.1 Low-rank Orthogonal Parameterization Scheme

The core premise of BRO revolves around a low-rank parameterization applied to an orthogonal layer, as introduced by the following proposition. A detailed proof is provided in Appendix A.1.

**Proposition 1.** *Let $V \in \mathbb{R}^{m \times n}$ be a matrix of rank $n$, and, without loss of generality, assume $m \geq n$. Then the parameterization $W = I - 2V(V^T V)^{-1} V^T$ satisfies the following properties:*

1. *$W$ is orthogonal and symmetric, i.e., $W^T = W$ and $W^T W = I$.*
2. *$W$ is an $n$-rank perturbation of the identity matrix, i.e., it has $n$ eigenvalues equal to $-1$ and $m - n$ eigenvalues equal to $1$.*
3. *$W$ degenerates to the negative identity matrix when $V$ is a full-rank square matrix.*

This parameterization draws inspiration from the block reflector (Dietrich, 1976; Schreiber & Parlett, 1988), which is widely used in parallel QR decomposition and is also important in other contemporary matrix factorization techniques. This approach enables the parameterization of an orthogonal matrix derived from a low-rank unconstrained matrix, thereby improving the computational efficiency.

Building on the definitive property of the proposition above, we initialize the parameter matrix $V$ as non-square to prevent it from degenerating into a negative identity matrix.

While the above discussion revolves around weight matrices for dense layers, the parameterization can also be used to construct orthogonal convolution operations. Specifically, given an unconstrained kernel $V \in \mathbb{R}^{c \times n \times k \times k}$, where each slicing $V_{:,:,i,j}$ is defined as in Proposition 1, $W_{\text{Conv}} = I_{\text{Conv}} - 2V \circledast (V^T \circledast V)^{-1} \circledast V^T$ constitutes a 2D multi-channel orthogonal convolution, where the operation $\circledast$ represents the convolution operation. Note that computing the inverse of a convolution kernel is challenging; therefore, we solve it in the Fourier domain instead. Following Cayley and LOT, we apply the 2D convolution theorem (Jain, 1989) to perform the convolution operation. Define FFT : $\mathbb{R}^{s \times s} \to \mathbb{C}^{s \times s}$ as the 2D Fourier transform operator and $\text{FFT}^{-1} : \mathbb{C}^{s \times s} \to \mathbb{C}^{s \times s}$ as its inverse, where $s \times s$ denotes the spatial dimensions, and the input will be zero-padded to $s \times s$ if the original shape is smaller. The 2D convolution theorem asserts that the circular convolution of two matrices in the spatial domain corresponds to their element-wise multiplication in the Fourier domain. Furthermore, based on the idea that multi-channel 2D circular convolution in the Fourier domain corresponds to a batch of matrix-vector products, we can perform orthogonal convolution as follows. Let $\tilde{X} = \text{FFT}(X)$ and $\tilde{V} = \text{FFT}(V)$, the convolution operation $Y = W_{\text{Conv}} \circledast X$ is then computed as $Y = \text{FFT}^{-1}(\tilde{Y})$ and $\tilde{Y}_{:,i,j} = \tilde{W}_{:,:,i,j} \tilde{X}_{:,i,j}$, where $\tilde{W}_{:,:,i,j} = I - 2\tilde{V}_{:,:,i,j}(\tilde{V}^*_{:,:,i,j} \tilde{V}_{:,:,i,j})^{-1} \tilde{V}^*_{:,:,i,j}$ and $i, j$ are the pixel indices. Note that the FFT is performed on the spatial (pixel) dimension, while the orthogonal multiplication is performed on the channel dimension.

**Algorithm 1** BRO Convolution Layer

1: **Input:** Tensor $X \in \mathbb{R}^{c \times s \times s}$, Kernel $V \in \mathbb{R}^{c \times n \times k \times k}$ with $n \leq c$ ▷ $c$ is channel size.
2: $X^{\text{pad}} := \text{zero\_pad}(X, (k, k, k, k)) \in \mathbb{R}^{c \times (s+2k) \times (s+2k)}$
3: $V^{\text{pad}} := \text{zero\_pad}(V, (0, 0, k + s, k + s)) \in \mathbb{R}^{c \times n \times (s+2k) \times (s+2k)}$
4: $\tilde{X} := \text{FFT}(X^{\text{pad}}) \in \mathbb{C}^{c \times (s+2k) \times (s+2k)}$ ; $\tilde{V} := \text{FFT}(V^{\text{pad}}) \in \mathbb{C}^{c \times n \times (s+2k) \times (s+2k)}$
5: **for all** $i, j \in \{1, \dots, s + 2k\}$ **do**
6: $\quad \tilde{Y}_{:,i,j} := (I - 2\tilde{V}_{:,:,i,j}(\tilde{V}_{:,:,i,j}^* \tilde{V}_{:,:,i,j})^{-1} \tilde{V}_{:,:,i,j}^*) \tilde{X}_{:,i,j}$ ▷ Apply our parameterization.
7: **end for**
8: **return** $(\text{FFT}^{-1}(\tilde{Y})_{:,k:-k,k:-k}).\text{real}$ ▷ Extract the real part.

**Proposition 2.** *Let* $\tilde{X} = FFT(X) \in \mathbb{C}^{c \times s \times s}$ *and* $\tilde{V} = FFT(V) \in \mathbb{C}^{c \times n \times s \times s}$, *the proposed BRO convolution* $Y = FFT^{-1}(\tilde{Y})$, *where* $\tilde{Y}_{:,i,j} = \tilde{W}_{:,:,i,j} \tilde{X}_{:,i,j}$ *and* $\tilde{W}_{:,:,i,j} = I - 2\tilde{V}_{:,:,i,j}(\tilde{V}_{:,:,i,j}^* \tilde{V}_{:,:,i,j})^{-1} \tilde{V}_{:,:,i,j}^*$, *is a real,* orthogonal *multi-channel 2D circular convolution.*

Importantly, the BRO convolution is a 2D circular convolution and is orthogonal, as demonstrated by Proposition 2. Furthermore, Proposition 2 guarantees that, although the BRO convolution primarily involves complex number computations in the Fourier domain, the output $Y$ remains real. The proof of Proposition 2 is provided in Appendix A.2. Additionally, an detailed proof of BRO's orthogonality is also included there.

Following LOT, we zero pad the input and parameters to size $s + 2k$, where $2k$ is the extra padding added to prevent circular convolution at the edges. A minor norm drop caused by removing the output padding is discussed in detail in Appendix A.4. Algorithm 1 details the proposed method, illustrating the case where the input and output channels are equal to $c$. For layers where the input dimension differs from the output dimension, we enforce the 1-Lipschitz constraint via semi-orthogonal matrices. In these matrices, only one side of the orthogonality condition is satisfied: either $W^T W = I$ or $W W^T = I$. We derive the parameterization of these matrices by first constructing an orthogonal matrix $W$ and then truncating it to the required dimensions. Specifically, for BRO convolution, let the input and output channel sizes be $c_{\text{in}}$ and $c_{\text{out}}$, respectively. Define $c = \max(c_{\text{out}}, c_{\text{in}})$. For each index $i$ and $j$, we parameterize $\tilde{V}_{:,:,i,j} \in \mathbb{C}^{c \times n}$ as $\tilde{W}_{:,:,i,j} \in \mathbb{C}^{c \times c}$, which is then truncated to $\tilde{W}_{:c_{\text{out}},:c_{\text{in}},i,j} \in \mathbb{C}^{c_{\text{out}} \times c_{\text{in}}}$. For details about the semi-orthogonal layer, refers to Appendix A.3.

### 4.2 PROPERTIES OF BRO LAYER

This section compares BRO to SOC and LOT, the state-of-the-art orthogonal layers.

**Iterative Approximation-Free** Both LOT and SOC utilize iterative algorithms for constructing orthogonal convolution layers. Although these methods' error bounds are theoretically proven to converge to zero, empirical observations suggest potential violations of the 1-Lipschitz constraint. Prior work (Béthune et al., 2022) has noted that SOC's construction may result in non-1-Lipschitz layers due to approximation errors inherent in the iterative process involving a finite number of terms in the Taylor expansion. Regarding LOT, we observe numerical instability during training due to the Newton method for orthogonal matrix computation. Specifically, the Newton method break the orthogonality

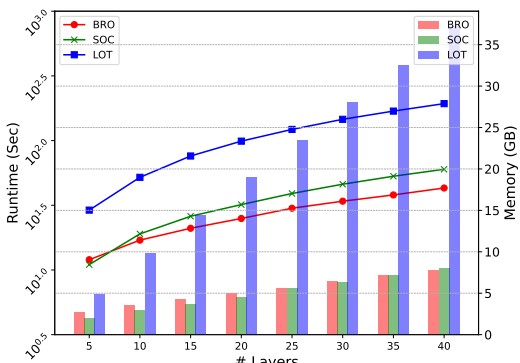

Figure 2: Comparison of runtime and memory usage among SOC, LOT, and BRO.

when encountering ill-conditioned parameters, even with the 64-bit precision computation recommended by the authors. An illustrative example is that using Kaiming initialization (He et al., 2015) instead of identity initialization results in a non-orthogonal layer. Detailed experiments are provided in Appendix D.6. In contrast, the proposed BRO constructs or-

thogonal layers without iterative approximation, ensuring both orthogonality and robustness certification validity.

**Time and Memory Efficiency** LOT's internal Newton method requires numerous steps to approximate the square root of the kernel, significantly prolonging training time and increasing memory usage. Conversely, the matrix operations in BRO are less complex, leading to substantially less training time and memory usage. Moreover, the low-rank parametrization characteristic of BRO further alleviates the demand for computational resources. When comparing BRO to SOC, BRO has an advantage in terms of inference time as SOC requires multiple convolution operations to compute the exponential convolution. Figure 2 shows the runtime per epoch and the memory usage during training. We analyze the computational complexity of different orthogonal layers both theoretically and empirically. The detailed comparison can be found in Appendix A.5.

**Non-universal Orthogonal Parameterization** While a single BRO layer is not a universal approximator for orthogonal layers, as established in the second property of Proposition 1, we empirically demonstrate in Section 7.2 that the expressive power of deep neural networks constructed using BRO is competitive with that of LOT and SOC.

## 5 BRONet Architecture

We design our architecture BRONet similar to SLL and LiResNet. It consists of a stem layer for image-to-feature conversion, several convolutional backbone blocks of same width for feature extraction, a neck block to convert feature maps into flattened vectors, and multiple dense blocks followed by a spectral normalized LLN layer (Singla et al., 2022). For non-linearity, MaxMin activation (Anil et al., 2019; Chernodub & Nowicki, 2016) is used. Further details can be found in Appendix B.2.

Compared to LiResNet with Lipschitz-regularized (Lip-reg) convolutional backbone blocks and SLL with SDP-based 1-Lipschitz layers, all the backbone blocks are BRO orthogonal parameterized, which ensures a tight Lipschitz composition bound in Equation 2 and is free from gradient norm vanishing. We keep the first stem layer in BRONet to be the only Lipschitz-regularized layer since we empirically find it benefits the model training with a more flexible Lipschitz control. Note that the Lipschitz composition bound of BRONet remains tight due to the orthogonal backbone blocks. Let the stem layer be $W_1$ and $Q$ be the composition of the layers before the neck block with $Q^T Q = I$, we have:

$$\text{Lip}(QW_1) = \sqrt{\lambda_{\max}((QW_1)^T(QW_1))} = \sqrt{\lambda_{\max}(W_1^T W_1)} = \text{Lip}(W_1), \tag{3}$$

where $\lambda_{\max}(\cdot)$ is the largest eigenvalue. Conversely, stacking multiple non-orthogonal layers such as Lip-reg or SLL does not necessarily results in a tight Lipschitz bound in Equation 2.

## 6 Logit Annealing Loss Function

Singla et al. (2022) posited that cross-entropy (CE) loss is inadequate for training Lipschitz models, as it fails to increase the margin. Thus, they integrated Certificate Regularization (CR) with the CE loss, formulated as: $\mathcal{L}_{\text{CE}} - \gamma \max(\mathcal{M}_f(x), 0)$, where $\mathcal{M}_f(x) = f(x)_t - \max_{k \neq t} f(x)_k$ is the logit margin between the ground-truth class $t$ and the runner-up class. $\gamma \max(\mathcal{M}_f(x), 0)$ is the CR term and $\gamma$ is a hyper-parameter. However, our investigation identifies several critical issues associated with the CR term, such as discontinuous loss gradient and gradient domination. Please see Appendix C.2 for details.

Our insight reveals that Lipschitz neural networks inherently possess limited model complexity, which impedes empirical risk minimization. Here, we utilize Rademacher complexity to justify that the empirical margin loss risk (Bartlett et al., 2017) is challenging to minimize with Lipschitz neural networks. Let $\mathcal{H}$ represent the hypothesis set. The empirical Rademacher complexity of $\mathcal{H}$ over a set $S = \{x_1, x_2, \ldots, x_n\}$ is given by:

$$\mathfrak{R}_S(\mathcal{H}) = \mathbb{E}_\sigma \left[ \sup_{h \in \mathcal{H}} \frac{1}{n} \sum_{i=1}^n \sigma_i h(x_i) \right], \tag{4}$$

where $\sigma_i$ are independent Rademacher variables uniformly sampled from $\{-1, 1\}$. Next, we use the Rademacher complexity to demonstrate that a model with low capacity results in a greater lower bound for margin loss risk.

**Theorem 1.** *Given a neural network $f$ and a set $S$ of size $n$, let $\ell_\tau$ denote the ramp loss (a special margin loss, see Appendix C) (Bartlett et al., 2017). Let $\mathcal{F}$ represent the hypothesis set of $f$. Define that:*

$$\mathcal{F}_\tau := \{(x, y) \mapsto \ell_\tau(\mathcal{M}(f(x), y))) : f \in \mathcal{F}\}; \tag{5}$$

$$\hat{\mathcal{R}}_\tau(f) := \frac{\sum_i \ell_\tau(\mathcal{M}(f(x_i), y_i))}{n}. \tag{6}$$

*Assume that $\mathcal{P}_e$ is the prediction error probability. Then, with probability $1 - \delta$, the empirical margin loss risk $\hat{\mathcal{R}}_\tau(f)$ is lower bounded by:*

$$\hat{\mathcal{R}}_\tau(f) \geq \mathcal{P}_e - 2\mathfrak{R}_S(\mathcal{F}_\tau) - 3\sqrt{\frac{\ln(1/\delta)}{2n}}. \tag{7}$$

Furthermore, for the $L$-Lipschitz neural networks, we introduce the following inequality to show that the model complexity is upper bounded by $L$.

**Proposition 3.** *Let $\mathcal{F}$ be the hypothesis set of the $L$-Lipschitz neural network $f$, and $\ell_\tau$ is the ramp loss with Lipschitz constant $1/\tau$, for some $\tau > 0$. Then, given a set $S$ of size $n$, we have:*

$$\mathfrak{R}_S(\mathcal{F}_\tau) = \mathbb{E}_\sigma \left[ \frac{1}{n} \sup_{f \in \mathcal{F}} \sum_{i=1}^n \sigma_i (\ell_\tau \circ f)(x_i) \right] \leq \frac{1}{\tau} \mathbb{E}_\sigma \left[ \frac{1}{n} \sup_{f \in \mathcal{F}} \sum_{i=1}^n \sigma_i f(x_i) \right] \leq \frac{L}{\tau \cdot n} \sum_{i=1}^n ||x_i||. \tag{8}$$

This is also known as Ledoux-Talagrand contraction (Ledoux & Talagrand, 2013). In Lipschitz neural networks, the upper bound is typically lower than in standard networks due to the smaller Lipschitz constant $L$, consequently limiting $\mathfrak{R}_S(\mathcal{F}_\tau)$.

According to Theorem 1, the empirical margin loss risk exhibits a greater lower bound if $\mathfrak{R}_S(\mathcal{F}_\tau)$ is low. It is important to note that the risk of the CR term, i.e., CR loss risk, is exactly the margin loss risk decreased by one unit when $\tau = 1/\gamma$. That is $\hat{\mathcal{R}}_{CR}(f) = \hat{\mathcal{R}}_\tau(f) - 1$. This indicates that CR loss risk also exhibits a greater lower bound. Thus, it is unlikely to minimize the CR term indefinitely if the model exhibits limited Rademacher complexity. Note that limited Rademacher complexity can result from a low Lipschitz constant or a large sample set. This also implies that we cannot limitlessly enlarge the margin in Lipschitz networks, especially for large real-world datasets. Detailed proofs can be found in Appendix C.

The CR term encourages a large margin for every data point simultaneously, which is impossible since the risk has a great lower bound. Due to the limited capacity of Lipschitz models, we must design a mechanism that enables models to learn appropriate margins for most data points. Specifically, when a data point exhibits a large margin, indicating further optimizing it is less beneficial, its loss should be annealed to allocate capacity for other data points. Based on this idea, we design a logit annealing mechanism to modulate the learning process, gradually reducing loss values of the large-margin data points. Consequently, we propose a novel loss function: the *Logit Annealing* (LA) loss. Let $\boldsymbol{z} = f(x)$ represent the logits output by the neural network, and let $\boldsymbol{y}$ be the one-hot encoding of the true label $t$. We define the LA loss as follows:

$$\mathcal{L}_{\text{LA}}(\boldsymbol{z}, \boldsymbol{y}) = -T(1 - \boldsymbol{p}_t)^\beta \log(\boldsymbol{p}_t), \text{where } \boldsymbol{p} = \text{softmax}(\tfrac{\boldsymbol{z} - \xi \boldsymbol{y}}{T}). \tag{9}$$

The hyper-parameters temperature $T$ and offset $\xi$ are adapted from the loss function in Prach & Lampert (2022) for margin training. The term $(1 - \boldsymbol{p}_t)^\beta$, referred to as the annealing mechanism, draws inspiration from Focal Loss (Lin et al., 2017). During training, LA loss initially promotes a moderate margin for each data point, subsequently annealing the data points with large margins as training progresses. Unlike the CR term, which encourages aggressive margin maximization, our method employs a balanced learning strategy that effectively utilizes the model's capacity, especially when it is limited. Consequently, LA loss allows Lipschitz models to learn an appropriate margin for most data points. Please see Appendix C for additional details on LA loss.

Table 1: Comparison of our method's performance with previous works. The $\ell_2$ perturbation budget $\varepsilon$ for certified accuracy is chosen following the convention of previous works. For fair comparison, diffusion-generated synthetic datasets are not used.

| Datasets | Models | #Param. | Clean Acc. | Cert. Acc. $(\varepsilon)$ | | |
|---|---|---|---|---|---|---|
| | | | | $\frac{36}{255}$ | $\frac{72}{255}$ | $\frac{108}{255}$ |
| CIFAR10 | Cayley Large (Trockman & Kolter, 2020) | 21M | 74.6 | 61.4 | 46.4 | 32.1 |
| | SOC-20 (Singla et al., 2022) | 27M | 76.3 | 62.6 | 48.7 | 36.0 |
| | LOT-20 (Xu et al., 2022) | 18M | 77.1 | 64.3 | 49.5 | 36.3 |
| | CPL XL (Meunier et al., 2022) | 236M | 78.5 | 64.4 | 48.0 | 33.0 |
| | AOL Large (Prach & Lampert, 2022) | 136M | 71.6 | 64.0 | 56.4 | **49.0** |
| | SOC-20+CRC (Singla & Feizi, 2022) | 40M | 79.1 | 66.5 | 52.5 | 38.1 |
| | SLL X-Large (Araujo et al., 2023) | 236M | 73.3 | 64.8 | 55.7 | 47.1 |
| | LiResNet (Hu et al., 2024) | 83M | 81.0 | 69.8 | 56.3 | 42.9 |
| | BRONet-M | 37M | 81.2 | 69.7 | 55.6 | 40.7 |
| | BRONet-L | 68M | **81.6** | **70.6** | **57.2** | 42.5 |
| CIFAR100 | Cayley Large (Trockman & Kolter, 2020) | 21M | 43.3 | 29.2 | 18.8 | 11.0 |
| | SOC-20 (Singla et al., 2022) | 27M | 47.8 | 34.8 | 23.7 | 15.8 |
| | LOT-20 (Xu et al., 2022) | 18M | 48.8 | 35.2 | 24.3 | 16.2 |
| | CPL XL (Meunier et al., 2022) | 236M | 47.8 | 33.4 | 20.9 | 12.6 |
| | AOL Large (Prach & Lampert, 2022) | 136M | 43.7 | 33.7 | 26.3 | 20.7 |
| | SOC-20+CRC (Singla & Feizi, 2022) | 40M | 51.8 | 38.5 | 27.2 | 18.5 |
| | SLL X-Large (Araujo et al., 2023) | 236M | 47.8 | 36.7 | 28.3 | **22.2** |
| | Sandwich (Wang & Manchester, 2023) | 26M | 46.3 | 35.3 | 26.3 | 20.3 |
| | LiResNet (Hu et al., 2024) | 83M | 53.0 | **40.2** | 28.3 | 19.2 |
| | BRONet-M | 37M | 54.1 | 40.1 | 28.5 | 19.6 |
| | BRONet-L | 68M | **54.3** | **40.2** | **29.1** | 20.3 |
| Tiny-ImageNet | SLL X-Large (Araujo et al., 2023) | 1.1B | 32.1 | 23.2 | 16.8 | 12.0 |
| | Sandwich (Wang & Manchester, 2023) | 39M | 33.4 | 24.7 | 18.1 | **13.4** |
| | LiResNet (Hu et al., 2024) | 133M | 40.9 | 26.2 | 15.7 | 8.9 |
| | BRONet | 75M | **41.2** | **29.0** | **19.0** | 12.1 |

# 7 EXPERIMENTS

In this section, we first evaluate the overall performance of our proposed BRONet against the $\ell_2$ certified robustness baselines. Next, to further demonstrate the effectiveness of the BRO layer, we conduct fair and comprehensive evaluations on multiple architectures for comparative analysis with orthogonal and other Lipschitz layers in previous literature. Lastly, we present the experimental results and analysis on the LA loss function. For detailed implementation information, refer to Appendix B.

## 7.1 MAIN RESULTS

We compare BRONet to the current leading methods in the literature. Figure 1 presents a visual comparison on CIFAR-10. Furthermore, Table 1 details the clean accuracy, certified accuracy, and the number of parameters. On CIFAR-10 and CIFAR-100, our model achieves the best clean and certified accuracy with the $\ell_2$ perturbation budget $\varepsilon = 36/255$. On the Tiny-ImageNet dataset, our method surpasses all baselines in terms of overall performance, demonstrating its scalability. Notably, BRONets achieves these results with a reasonable number of parameters.

## 7.2 ABLATION STUDIES

**Extra Diffusion Data Augmentation**  As demonstrated in previous studies (Hu et al., 2024; Wang et al., 2023), incorporating additional synthetic data generated by diffusion models such as elucidating diffusion model (EDM) (Karras et al., 2022) can enhance performance. We evaluate the effectiveness of our method in this setting, using diffusion-generated synthetic datasets from Hu et al. (2024); Wang et al. (2023) for CIFAR-10 and CIFAR-100, which contain post-filtered 4 million and 1 million images, respectively. Table 2 presents the results,

Table 2: Improvements of combining LA and BRO with LiResNet using diffusion data augmentation. The best results of each dataset are marked in bold. Performance improvements and degradations relative to the baseline are marked in green and red, respectively.

| Datasets | Methods | Clean Acc. | Cert. Acc. ($\varepsilon$) | | |
|---|---|---|---|---|---|
| | | | $\frac{36}{255}$ | $\frac{72}{255}$ | $\frac{108}{255}$ |
| CIFAR-10 (+EDM) | LiResNet | 87.0 | 78.1 | 66.1 | 53.1 |
| | +LA | 86.7 (-0.3) | 78.1 (+0.0) | 67.0 (+0.9) | 54.2 (+1.1) |
| | +LA + BRO | **87.2** (+0.2) | **78.3** (+0.2) | **67.4** (+1.3) | **54.5** (+1.4) |
| CIFAR-100 (+EDM) | LiResNet | 61.0 | 48.4 | 36.9 | 26.5 |
| | +LA | 61.1 (+0.1) | 48.9 (+0.5) | 37.5 (+0.6) | **27.6** (+1.1) |
| | +LA + BRO | **61.6** (+0.6) | **49.1** (+0.7) | **37.7** (+0.8) | 27.2 (+0.7) |

Table 3: Comparison of clean and certified accuracy using different Lipschitz convolutional backbones. The best results are marked in bold. #Layers is the number of convolutional backbone layers, and #param. is the number of parameters in the constructed architecture.

| Conv. Backbone | #Layers | #Param. | CIFAR-10 (+EDM) | | | | CIFAR-100 (+EDM) | | | |
|---|---|---|---|---|---|---|---|---|---|---|
| | | | Clean | $\frac{36}{255}$ | $\frac{72}{255}$ | $\frac{108}{255}$ | Clean | $\frac{36}{255}$ | $\frac{72}{255}$ | $\frac{108}{255}$ |
| LOT | 2 | 59M | 85.7 | 76.4 | 65.1 | 52.2 | 59.4 | 47.6 | 36.6 | 26.3 |
| Cayley | 6 | 68M | 86.7 | 77.7 | 66.9 | 54.3 | 61.1 | 48.7 | **37.8** | 27.5 |
| Cholesky | 6 | 68M | 85.4 | 76.6 | 65.7 | 53.3 | 59.4 | 47.4 | 36.8 | 26.9 |
| SLL | 12 | 83M | 85.6 | 76.8 | 66.0 | 53.3 | 59.4 | 47.6 | 36.6 | 27.0 |
| SOC | 12 | 83M | 86.6 | 78.2 | 67.0 | 54.1 | 60.9 | 48.9 | 37.6 | **27.8** |
| Lip-reg | 12 | 83M | 86.7 | 78.1 | 67.0 | 54.2 | 61.1 | 48.9 | 37.5 | 27.6 |
| BRO | 12 | 68M | **87.2** | **78.3** | **67.4** | **54.5** | **61.6** | **49.1** | 37.7 | 27.2 |

showing that combining LA and BRO effectively leverages these synthetic datasets to enhance performance.

**Backbone Comparison** As the improvements in the previous work by Hu et al. (2024) primarily stem from using diffusion-generated synthetic datasets and architectural changes, we conduct a fair and comprehensive comparison of different Lipschitz convolutional layers using the default LiResNet architecture (with Lipschitz-regularized convolutional layers), along with LA and diffusion-based data augmentation. The only modification is swapping out the convolutional backbone layers. It is important to note that for FFT-based orthogonal layers (excluding BRO), we must reduce the number of backbone layers to stay within memory constraints. LOT has the fewest parameters due to its costly parameterization. With half-rank parameterization in BRO, the number of parameters for BRO, Cayley, and Cholesky remain consistent, while SLL, SOC, and Lipschitz-regularized retain the original number of parameters. The results in Table 3 indicate that BRO is the optimal backbone choice compared to other layers in terms of overall performance.

**LipConvNet Benchmark** To further validate the effectiveness of BRO, we also evaluate it on LipConvNets, which have been the standard architecture in the literature on orthogonal layers. For LipConvNets details, refer to Appendix B.2. Table 4 illustrates the certified robustness of SOC, LOT, and BRO layers. It is evident that the LipConvNet constructed by BRO layers compares favorably to the other orthogonal layers in terms of clean and robust accuracy. Detailed comparisons are provided in Appendix D.5.

### 7.3 LA Loss Effectiveness

Table 2 illustrates the performance improvements achieved using the proposed LA loss, with LA showing better results on CIFAR-100 compared to CIFAR-10. We also provide extensive ablation experiments in Appendix D.3 to validate its effectiveness on different architectures and datasets. Our experiments show that LA loss promotes a balanced margin, increasing clean and certified robust accuracies by approximately 1% to 2%, especially for models trained on more challenging datasets like Tiny-ImageNet. See Table 11 for more details.

Table 4: Comparison of clean and certified accuracy with different orthogonal layers in LipConvNets (depth-width). Instances marked with a dash (-) indicate out of memory during training. The best results with each model are marked with bold.

| Models | Layers | CIFAR-100 | | | | Tiny-ImageNet | | | |
|---|---|---|---|---|---|---|---|---|---|
| | | Clean | $\frac{36}{255}$ | $\frac{72}{255}$ | $\frac{108}{255}$ | Clean | $\frac{36}{255}$ | $\frac{72}{255}$ | $\frac{108}{255}$ |
| LipConvNet (10-32) | SOC | 47.5 | 34.7 | 24.0 | 15.9 | 38.0 | 26.5 | 17.7 | 11.3 |
| | LOT | **49.1** | **35.5** | 24.4 | **16.3** | **40.2** | 27.9 | **18.7** | **11.8** |
| | BRO | 48.6 | 35.4 | **24.5** | 16.1 | 39.4 | **28.1** | 18.2 | 11.6 |
| LipConvNet (10-48) | SOC | 48.2 | 34.9 | 24.4 | 16.2 | 38.9 | 27.1 | 17.6 | 11.2 |
| | LOT | **49.4** | 35.8 | 24.8 | 16.3 | - | - | - | - |
| | BRO | **49.4** | **36.2** | **24.9** | **16.7** | **40.0** | **28.1** | **18.9** | **12.3** |
| LipConvNet (10-64) | SOC | 48.5 | 35.5 | 24.4 | 16.3 | 39.3 | 27.3 | 17.6 | 11.2 |
| | LOT | 49.6 | 36.1 | 24.7 | 16.2 | - | - | - | - |
| | BRO | **49.7** | **36.7** | **25.2** | **16.8** | **40.7** | **28.4** | **19.2** | **12.5** |

To demonstrate that the LA loss enables learning an appropriate margin for most data points, we further investigate the certified radius distribution. Following Cohen et al. (2019), we plot the certified accuracy with respect to the radius on CIFAR-100 to visualize the margin distribution in Figure 3. The certified radius is proportional to the margin in Lipschitz models. Thus, the x-axis and y-axis can be seen as margin and complementary cumulative distribution of data points, respectively (Lecuyer et al., 2019). The results indicate that the number of data points with appropriate margins increases, which is evident as the red curve rises higher than the others at the radius between $[0.0, 0.6]$. Moreover, the clean accuracy, which corresponds to certified accuracy at zero radius, is also observed to be slightly higher. This suggests that the LA loss does not compromise clean accuracy for robustness. To further understand the annealing mechanism, we analyze the distribution of the certified radius across the data points, as shown in Table 5. Compared to CR, the LA loss reduces both the positive skewness and variance of the distribution, indicating a rightward shift in the peak and a decrease in the dispersion of the radius. This suggests that LA loss helps mitigate the issue of overfitting to certain data points and improves the certified radius for most points. Additional experiments, including an empirical robustness test and ablation studies on BRO rank and loss, are presented in Appendix D.

Table 5: The median, variance and skewness of certified radius distribution.

| Loss | Median | Variance | Skewness |
|---|---|---|---|
| CE | 0.2577 | 0.0732 | 1.2114 |
| CE+CR | 0.2750 | 0.1000 | 1.4843 |
| LA | 0.2840 | 0.0797 | 1.0539 |

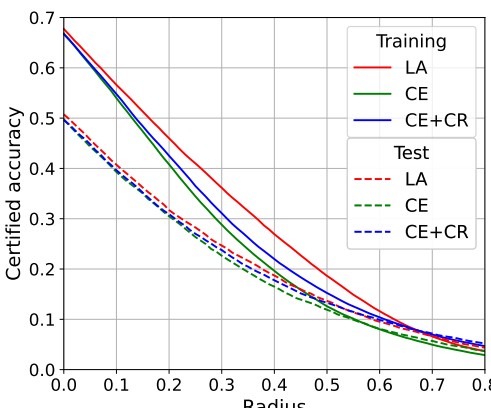

Figure 3: Certified accuracy with respect to radius. LA loss helps learn appropriate margin.

## 8 Conclusion

In this paper, we introduce a novel BRO layer to construct various Lipschitz neural networks. The BRO layer features low-rank parameterization and is free from iterative approximations. As a result, it is both memory and time efficient compared with existing orthogonal layers, making it well-suited for integration into advanced Lipschitz architectures to enhance robustness. Furthermore, extensive experimental results have shown that BRO is one of the most promising orthogonal convolutional layers for constructing expressive Lipschitz networks. Next, we address the limited complexity issue of Lipschitz neural networks and introduce the new Logit Annealing loss function to help models learn appropriate margins. Moving forward, the principles and methodologies in this paper could serve as a foundation for future research in certifiably robust network design.

## Reproducibility

To ensure the reproducibility of our experiments, we have provided detailed implementation of the proposed BRO method. Additionally, the code is included in the supplementary material, enabling readers to replicate the experiments. The implementation of Algorithm 1 can be found in `lipconvnet/models/layers/bro_conv.py`.

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

## A BRO Layer Analysis

### A.1 Proof of Proposition 1

**Proposition 1.** *Let $V \in \mathbb{R}^{m \times n}$ be a matrix of rank $n$, and, without loss of generality, assume $m \geq n$. Then the parameterization $W = I - 2V(V^TV)^{-1}V^T$ satisfies the following properties:*

1. *$W$ is orthogonal and symmetric, i.e., $W^T = W$ and $W^TW = I$.*

2. *$W$ is an n-rank perturbation of the identity matrix, i.e., it has $n$ eigenvalues equal to $-1$ and $m - n$ eigenvalues equal to 1.*

3. *$W$ degenerates to the negative identity matrix when $V$ is a full-rank square matrix.*

*Proof.* Assuming $V$ is as defined in Proposition 1, the symmetry of this parameterization is straightforward to verify. The orthogonality of $W$, however, requires confirmation that the following condition is satisfied:

$$
\begin{aligned}
WW^T &= (I - 2V(V^TV)^{-1}V^T)(I - 2V(V^TV)^{-1}V^T)^T \\
&= (I - 2V(V^TV)^{-1}V^T)(I - 2V(V^TV)^{-1}V^T) \\
&= I - 4V(V^TV)^{-1}V^T + 4V(V^TV)^{-1}V^TV(V^TV)^{-1}V^T \\
&= I - 4V(V^TV)^{-1}V^T + 4V(V^TV)^{-1}V^T \\
&= I.
\end{aligned}
\tag{10}
$$

Next, define $S = \{v_1, v_2, \cdots, v_n\}$ as the set of column vectors of $V$. Let $e_i$ denote the $i$-th standard basis vector in $\mathbb{R}^n$. Then, we have

$$
\begin{aligned}
Wv_i &= (I - 2V(V^TV)^{-1}V^T)v_i \\
&= v_i - 2V(V^TV)^{-1}V^Tv_i \\
&= v_i - 2V(V^TV)^{-1}(V^TVe_i) \\
&= v_i - 2V(V^TV)^{-1}(V^TV)e_i \\
&= v_i - 2Ve_i \\
&= v_i - 2v_i = -v_i.
\end{aligned}
\tag{11}
$$

For the vectors in the orthogonal complement of $S$, denoted by $S^\perp = \{v_{n+1}, v_{n+2}, \cdots, v_m\}$, we have

$$
Wv_i = (I - 2V(V^TV)^{-1}V^T)v_i = v_i.
\tag{12}
$$

The equality holds because, for all $v_i \in S^\perp$, we have $V^Tv_i = 0$.

Therefore, the eigenspace corresponding to eigenvalue $-1$ is spanned by $S$, while the eigenspace corresponding to eigenvalue 1 is spanned by $S^\perp$.

Assume $V$ is a full-rank square matrix, which implies that $V$ is invertible. Thus:

$$
\begin{aligned}
W &= I - 2V(V^TV)^{-1}V^T \\
&= I - 2VV^{-1}(V^T)^{-1}V^T \\
&= I - 2I \\
&= -I.
\end{aligned}
\tag{13}
$$

Thus, the proof is complete. $\qquad \square$

## A.2 PROOF OF PROPOSITION 2

★ This subsection has been fully revised. ★

**Proposition 2.** *Let* $\tilde{X} = FFT(X) \in \mathbb{C}^{c \times s \times s}$ *and* $\tilde{V} = FFT(V) \in \mathbb{C}^{c \times n \times s \times s}$, *the proposed BRO convolution* $Y = FFT^{-1}(\tilde{Y})$, *where* $\tilde{Y}_{:,i,j} = \tilde{W}_{:,:,i,j}\tilde{X}_{:,i,j}$ *and* $\tilde{W}_{:,:,i,j} = I - 2\tilde{V}_{:,:,i,j}(\tilde{V}^*_{:,:,i,j}\tilde{V}_{:,:,i,j})^{-1}\tilde{V}^*_{:,:,i,j}$, *is a real, orthogonal multi-channel 2D circular convolution.*

*Proof.* To establish the results of Proposition 2, we first present several supporting lemmas.

**Lemma 1.** *(2D convolution theorem) Let* $X, W \in \mathbb{R}^{s \times s}$, *and* $F \in \mathbb{R}^{s \times s}$ *be the DFT matrix. Then,* $\tilde{X} = FFT(X) = FXF$, *i.e., the DFT is applied to the rows and columns of* $X$. *In addition, let the 2D circular convolution of* $X$ *with* $W$ *be* $\mathsf{conv}_W(X) \in \mathbb{R}^{s \times s}$. *It follows that*

$$\tilde{W} \odot \tilde{X} = FWF \odot FXF = F\mathsf{conv}_W(X)F,$$

*where* $\odot$ *is the element-wise product.*

Next, we introduce the multi-channels 2D circular convolution. Following Trockman & Kolter (2020), we flatten the four-dimension tensors into matrices to facilitate the analysis. Let an input image with $c_{\mathsf{in}}$ input channels represent $X \in \mathbb{R}^{c_{\mathsf{in}} \times s \times s}$, it can be vectorized into $\mathcal{X} = [\mathsf{vec}^T(X_1), ..., \mathsf{vec}^T(X_{c_{\mathsf{in}}})]^T \in \mathbb{R}^{c_{\mathsf{in}}s^2}$. Similarly, the vectorized output is $\mathcal{Y} = [\mathsf{vec}^T(Y_1), ..., \mathsf{vec}^T(Y_{c_{\mathsf{out}}})]^T \in \mathbb{R}^{c_{\mathsf{out}}s^2}$. Then, we have a 2D circular convolution operation with $\mathcal{C} \in \mathbb{R}^{c_{\mathsf{out}}s^2 \times c_{\mathsf{in}}s^2}$ such that $\mathcal{Y} = \mathcal{C}\mathcal{X}$. Note that $\mathcal{C}$ has $c_{\mathsf{out}} \times c_{\mathsf{in}}$ blocks with size $s^2 \times s^2$.

**Lemma 2.** *(Trockman & Kolter, 2020, Corollary A.1.1) If* $\mathcal{C} \in \mathbb{R}^{c_{\mathsf{out}}s^2 \times c_{\mathsf{in}}s^2}$ *represents a 2D circular convolution with* $c_{\mathsf{in}}$ *input channels and* $c_{\mathsf{out}}$ *output channels, then it can be block diagonalized as*

$$\mathcal{F}_{c_{\mathsf{out}}}\mathcal{C}\mathcal{F}^*_{c_{\mathsf{in}}} = \mathcal{D}, \tag{14}$$

*where* $\mathcal{F}_c = S_{c,s^2}(I_c \otimes (F \otimes F))$, $S_{c,s^2}$ *is a permutation matrix,* $I_k$ *is the identity matrix of order* $k$, *and* $\mathcal{D}$ *is block diagonal with* $s^2$ *blocks of size* $c_{\mathsf{out}} \times c_{\mathsf{in}}$.

**Lemma 3.** *Consider* $J \in \mathbb{C}^{p \times p}$ *as a unitary matrix. Define* $V$ *and* $\tilde{V}$ *such that* $V = J\tilde{V}J^*$, *where* $V \in \mathbb{R}^{p \times p}$ *and* $\tilde{V} \in \mathbb{C}^{p \times p}$. *Let* $\mathsf{BRO}(V) = I - 2V(V^*V)^{-1}V^*$ *be our parameterization. Then,*

$$\mathsf{BRO}(V) = J\mathsf{BRO}(\tilde{V})J^*. \tag{15}$$

*Proof.* Assume $J$ and $V$ are as defined in Proposition 3. Then

$$\begin{aligned} J^*\mathsf{BRO}(V)J &= J^*(I - 2V(V^TV)^{-1}V^T)J \\ &= I - 2(J^*J\tilde{V}J^*)[(J\tilde{V}^*\tilde{V}J^*)]^{-1}(J\tilde{V}^*J^*J) \\ &= I - 2(\tilde{V}J^*)[(J\tilde{V}^*\tilde{V}J^*)]^{-1}(J\tilde{V}^*) \\ &= I - 2\tilde{V}(\tilde{V}^*\tilde{V})^{-1}\tilde{V}^* \qquad\qquad (\star) \\ &= \mathsf{BRO}(\tilde{V}). \end{aligned}$$

The equality at $(\star)$ holds because

$$(\tilde{V}^*\tilde{V})^{-1} = J^*(J\tilde{V}^*\tilde{V}J^*)^{-1}J.$$

□

We begin the proof under the assumption that the number of input channels equals the number of output channels, i.e., $c_{\mathsf{in}} = c_{\mathsf{out}} = c$. According to Lemma 2, the stacked weight matrix $\mathcal{C} \in \mathbb{R}^{c_{\mathsf{out}}s^2 \times c_{\mathsf{in}}s^2}$ can be diagonalized as follows:

$$\mathcal{C} = \mathcal{F}^*_c\mathcal{D}\mathcal{F}_c. \tag{16}$$

where $\mathcal{F}^*_c$ and $\mathcal{F}_c$ are unitary matrices, and $\mathcal{D}$ is a block diagonal matrix.

Note that since $\mathcal{D}$ is block diagonal, the BRO transformation of $\mathcal{D}$ can be expressed as:

$$\mathsf{BRO}(\mathcal{D}) = \mathsf{BRO}(\mathcal{D}_1) \oplus \mathsf{BRO}(\mathcal{D}_2) \oplus \cdots \oplus \mathsf{BRO}(\mathcal{D}_{s^2}),$$

where $\oplus$ denotes the direct sum. This is because each block $\mathcal{D}_k$ for $k = 1, \ldots, s^2$ is independently transformed by the BRO operation. Additionally, because the original weight matrix $\mathcal{C}$ is real, the BRO convolution $\mathsf{BRO}(\mathcal{C})$ remains real as well.

Applying Lemma 3 on Equation 16, we consider a real vectorized input $\mathcal{X}$. The output of the BRO convolution is given by:

$$\mathcal{Y} = \mathsf{BRO}(\mathcal{C})\mathcal{X} = \mathcal{F}_c^* \mathsf{BRO}(\mathcal{D})\mathcal{F}_c\mathcal{X}. \tag{17}$$

This ensures that $\mathcal{Y}$ is real. Consequently, Algorithm 1 is guaranteed to produce a real output when given a real input $\mathcal{X}$.

Finally, the orthogonality of the BRO convolution operation $\mathsf{BRO}(\mathcal{C})$ can be derived as follows. Since both $\mathcal{F}_c^*$ and $\mathcal{F}_c$ are unitary matrices (Trockman & Kolter, 2020), and $\mathsf{BRO}(\mathcal{D})$ is unitary as well, the composition of these unitary operations preserves orthogonality.

Thus, we have established that the BRO convolution operation $\mathsf{BRO}(\mathcal{C})$ is orthogonal, thereby completing the proof of Proposition 2.

$\square$

### A.3 ANALYSIS OF SEMI-ORTHOGONAL LAYER

★ This is a newly added subsection. ★

In this section, we provide the detailed analysis about semi-orthogonal BRO layers, which can be categorized into two types: dimension expanding layers and dimension reduction layers. To facilitate understanding, we begin with the dense version of BRO (a single 2D matrix).

For a expanding layer constructed with $W \in \mathbb{R}^{d_{\text{out}} \times d_{\text{in}}}$, where $d_{\text{in}} < d_{\text{out}}$, it satisfies the condition $W^T W = I_{d_{\text{in}}}$. Since the condition is equivalent to ensure that the columns are orthonormal, the norm of a vector is preserved when projecting onto its column space, which means $\|Wx\| = \|x\|$ for every $x \in \mathbb{R}^{d_{\text{in}}}$, thus, ensures 1-Lipschitz property.

For a reduction layer constructed with $W \in \mathbb{R}^{d_{\text{out}} \times d_{\text{in}}}$, where $d_{\text{in}} > d_{\text{out}}$, it satisfies the condition $WW^T = I_{d_{\text{out}}}$. Unlike expanding layers, the columns of $W$ in reduction layers cannot be orthonormal due to the dimensionality constraint $d_{\text{in}} > d_{\text{out}}$. Consequently, we have the following relationship for every $x \in \mathbb{R}^{d_{\text{in}}}$, $\|Wx\| \leq \|x\|$. The equality holds if $x$ lies entirely within the subspace spanned by the rows of $W$. In general, reduction layers do not preserve the norm of input vectors. However, they remain 1-Lipschitz bounded, ensuring that the transformation does not amplify the input norm.

The same principles apply to BRO convolution. The primary difference between the dense and convolution versions of BRO layers arises from the dimensions they are applied to. Therefore, the results discussed for BRO dense layers also hold for BRO convolution layers. Figure 4 visualizes BRO convolution under three different dimensional settings, illustrating the behavior of channel-expanding and channel-reduction operations in convolutional contexts.

### A.4 THE EFFECT OF ZERO-PADDING

★ This is a newly added subsection. ★

Following LOT (Xu et al., 2022), we apply zero-padding on images $X \in \mathbb{R}^{c \times s \times s}$, creating $X_{\text{pad}} \in \mathbb{R}^{c \times (s+2k) \times (s+2k)}$, before performing the 2D FFT. After applying $\text{FFT}^{-1}$, we obtain $Y_{\text{pad}} \in \mathbb{R}^{c \times (s+2k) \times (s+2k)}$, from which the padded pixels are removed to restore the original dimensions of $X$, resulting in $Y \in \mathbb{R}^{c \times s \times s}$. This approach leverages zero-padding to avoid

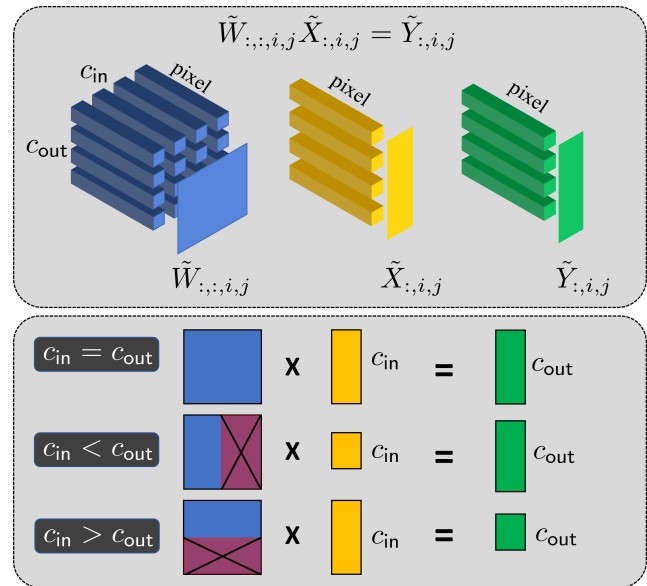

Figure 4: Visualization of BRO convolution for different $c_{\text{in}}$ and $c_{\text{out}}$.

circular convolution across edges, which empirically improves performance. However, since circular convolution is still applied to the entire image, including the padded regions, these padded areas can acquire information from the central spatial region. Moreover, as norm preservation only holds for $||X|| = ||X_{\text{pad}}|| = ||Y_{\text{pad}}||$, removing pixels from $Y_{\text{pad}}$ causes the slight norm drop. Importantly, it does not affect the validity of the certified results, as neither zero-padding nor the removal of padded parts expands the norm or violates the 1-Lipschitz bound.

### A.5 Complexity Comparison of Orthogonal Layers

In this section, we demonstrate the computational and memory advantages of the proposed method by analyzing its complexity compared to prior work. We use conventional notation from Prach et al. (2023). We focus on algorithmic complexity and required memory, particularly in terms of *multiply-accumulate operations (MACs)*. The detailed complexity comparison is presented in Table 7.

The analysis has two objectives: input transformation and parameter transformation. The computational complexity and memory requirements of the forward pass during training are the sum of the respective MACs and memory needs. The backward pass has the same complexity and memory requirements, increasing the overall complexity by a constant factor. In addition to theoretical complexity, we report the practical time and memory usage for different orthogonal layers under various settings in Figure 5.

In the following analysis, we consider only dimension-preserving layers, where the input and output channels are equal, denoted by $c$. Define the input size as $s \times s \times c$, the batch size as $b$, the kernel size as $k \times k$, the number of inner iterations of a method as $t$, and the rank-control factor for BRO as $\kappa$, as listed in Table 6. To simplify the analysis, we assume $c > \log_2(s)$. Under the PyTorch (Paszke et al., 2019) framework, we can also assume that rescaling a tensor by a scalar and adding two tensors do not require extra memory during back-propagation.

**Standard Convolution** In standard convolutional layers, the computational complexity of the input transformation is $C = bs^2 c^2 k^2$ MACs, and the memory requirement for input and kernel are $M = bs^2 c$ and $P = c^2 k^2$, respectively. Additionally, these layers do not require any computation for parameter transformation.

**SOC** For the SOC layer, $t$ convolution iterations are required. Thus, the input transformation requires computation complexity and memory $t$ times that of standard convolution. For the

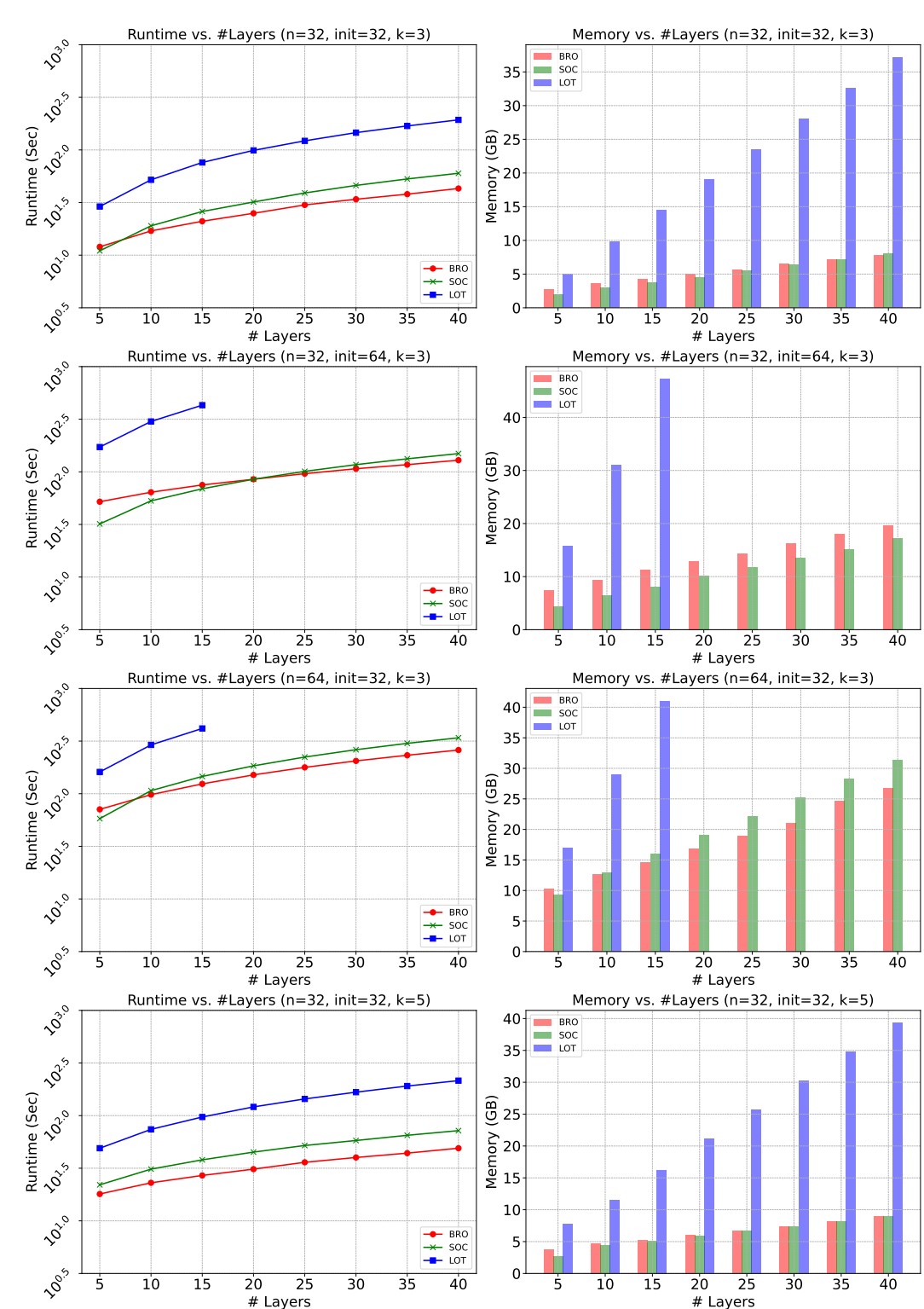

Figure 5: Demonstration of the runtime and memory consumption under different settings with LipConvNet architecture. The notation $n$ denotes the input size, init denote the initial channel of the the entire model, and $k$ denotes the kernel size. The batch sizes are fixed at 512 for all plots, and each value is the average over 10 iterations.

Table 6: Notation used in this section.

| Notation | Description |
|----------|-------------|
| $b$ | batch size |
| $k$ | kernel size |
| $c$ | input/output channels |
| $s$ | input size (resolution) |
| $t$ | number of internal iterations |
| $\kappa$ | Rank-Control factor for BRO |

Table 7: Computational complexity and memory requirements of different methods. We report multiply-accumulate operations (MACS) as well as memory requirements for batch size $b$, input size $s \times s \times c$, kernel size $k \times k$ and number of inner iterations $t$ for SOC and LOT, rank-control factor $\kappa \in [0, 1]$ for BRO. We denote the complexity and memory requirement of standard convolution as $C = bs^2c^2k^2$, $M = bs^2c$, and $P = c^2k^2$, respectively.

| Method | Input Transformations | | Parameter Transformations | |
|--------|------------------------|------------|----------------------------|------------------|
| | MACS $\mathcal{O}(\cdot)$ | Memory | MACS $\mathcal{O}(\cdot)$ | Memory $\mathcal{O}(\cdot)$ |
| Standard | $C$ | $M$ | - | $P$ |
| SOC | $Ct$ | $Mt$ | $c^2k^2t$ | $P$ |
| LOT | $bs^2c^2$ | $3M$ | $4s^2c^3t$ | $4s^2c^2t$ |
| BRO | $bs^2c^2$ | $2.5M$ | $s^2c^3\kappa$ | $2s^2c^2$ |

parameter transformation, a kernel re-parameterization is needed to ensure the Jacobian of the induced convolution is skew-symmetric. During training, the SOC layer applies Fantastic Four (Singla & Feizi, 2021a) technique to bound the spectral norm of the convolution, which incurs a cost of $c^2k^2t$. The memory consumption remains the same as standard convolution.

**LOT** The LOT layer achieves orthogonal convolution via Fourier domain operations. Applying the Fast Fourier Transform (FFT) to inputs and weights has complexities of $\mathcal{O}(bcs^2 \log(s^2))$ and $\mathcal{O}(c^2s^2 \log(s^2))$, respectively. Subsequently, $s^2$ matrix orthogonalizations are required using the transformation $V(V^TV)^{-\frac{1}{2}}$. The Newton Method is employed to find the inverse square root. Specifically, let $Y_0 = V^TV$ and $Z_0 = I$, then $Y_i$ is defined as

$$Y_{i+1} = \frac{1}{2}Y_i\left(3I - Z_iY_i\right), \quad Z_{i+1} = \frac{1}{2}\left(3I - Z_iY_i\right)Z_i. \tag{18}$$

This iteration converges to $(V^TV)^{-\frac{1}{2}}$. Executing this procedure involves computing $4s^2t$ matrix multiplications, requiring about $4s^2c^3t$ MACs and $4s^2c^2t$ memory. The final steps consist of performing $\frac{1}{2}bs^2$ matrix-vector products, requiring $\frac{1}{2}bs^2c^2$ MACs, as well as the inverse FFT. Given our assumption that $c > \log(s^2)$, the FFT operation is dominated by other operations. Considering the memory consumption, LOT requires padding the kernel from a size of $c \times c \times k \times k$ to $c \times c \times s \times s$, requiring $bs^2c^2$ memory. Additionally, we need to keep the outputs of the FFT and the matrix multiplications in memory, requiring about $4s^2c^2t$ memory each.

**BRO** Our proposed BRO layer also achieves orthogonal convolution via Fourier domain operations. Therefore, the input transformation requires the same computational complexity as LOT. However, by leveraging the symmetry properties of the Fourier transform of a real matrix, we reduce both the memory requirement and computational complexity by half. During the orthogonalization process, only $\frac{1}{2}s^2$ are addressed. The low-rank parameterization results in a complexity of approximately $s^2c^3\kappa$ and memory usage of $\frac{1}{2}s^2c^2$. Additionally, we need to keep the outputs of the FFT, the matrix inversion, and the two matrix multiplications in memory, requiring about $\frac{1}{2}s^2c^2t$ memory each.

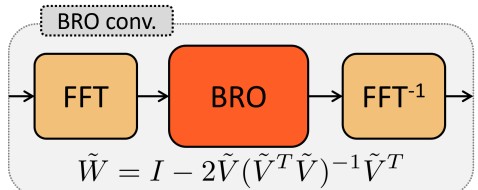

Figure 6: The proposed Block Reflector Orthogonal (BRO) convolution kernel, which is an orthogonal matrix, employs Fourier transformation to simulate the convolution operation. This convolution is inherently orthogonal and thus 1-Lipschitz, providing guarantees for adversarial robustness.

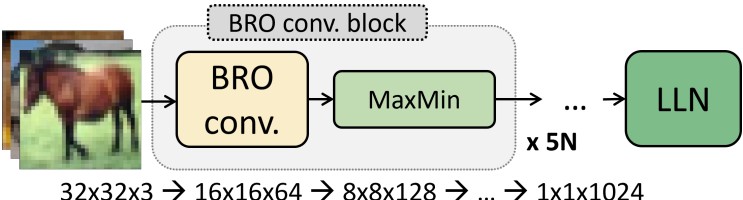

32x32x3 → 16x16x64 → 8x8x128 → ... → 1x1x1024

Figure 7: Following Trockman & Kolter (2020); Singla & Feizi (2021b); Xu et al. (2022), we use the proposed orthogonal convolution layer to construct the Lipschitz neural network. This figure illustrates the LipConvnet-5, which cascades five BRO convolution layers. The activation function used is the MaxMin function, and the final layer is the last layer normalization (LLN).

## B  IMPLEMENTATION DETAILS

In this section, we will detail our computational resources, the architectures of BRONet and LipConvNet, rank-n configuration, hyper-parameters used in LA loss, and experimental settings.

### B.1  COMPUTATIONAL RESOURCES

All experiments are conducted on a computer with an Intel Xeon Gold 6226R processor and 192 GB of DRAM memory. The GPU we used is the NVIDIA RTX A6000 (10,752 CUDA cores, 48 GB memory per card). For CIFAR-10 and CIFAR-100, we used a single A6000 card for training. For Tiny-ImageNet and diffusion data augmentation on CIFAR-10/100, we utilized distributed data parallel (DDP) across two A6000 cards for joint training. Training a LipConvNet-10 on this setup, as detailed in Table 12, required approximately 3,400 seconds.

### B.2  ARCHITECTURE DETAILS

The proposed BRO layer is illustrated in Figure 6. In this paper, we mainly use the BRO layer to construct two different architectures: BRONet and LipConvNet. We will first explain the details of BRONet, followed by an explanation of LipConvNet constructed using the BRO layer.

**BRONet Architecture** Figure 8 illustrates the details of the BRONet architecture, which is comprised of several key components:

- Stem: This consists of a unconstrained convolutional layer that is Lipschitz-regularized during training. The width $W$ is the feature channel dimension, which is an adjustable parameter.
- Backbone: This segment includes $L$ BRO convolutional blocks of channel width $W$, each adhering to the 1-Lipschitz constraint.

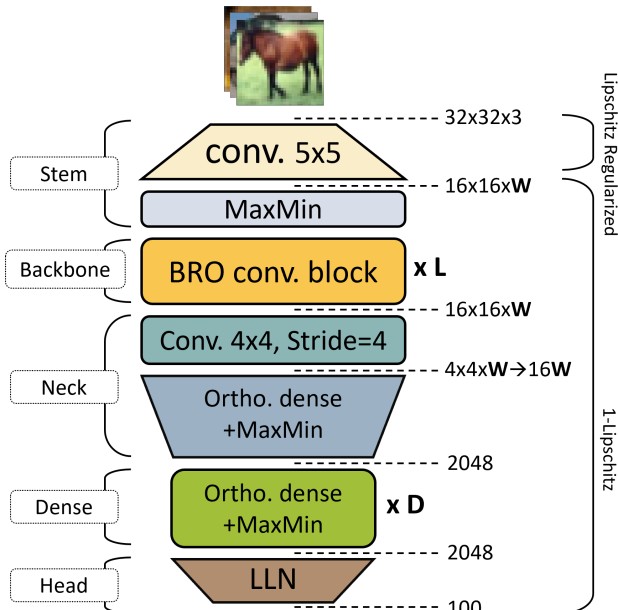

Figure 8: Following the LiResNet architecture (Leino et al., 2021; Hu et al., 2023), we utilized the BRO layer to construct **BRONet**. The parameters $L$, $W$, and $D$ can be adjusted to control the model size.

- Neck: This consists of a convolutional down-sampling layer followed by a dense layer, which reduces the feature dimension. For the convolutional layer, we follow LiResNet (Hu et al., 2024) to construct a 1-Lipschitz matrix with dimension $(c_{out}, c_{in} \times k^2)$ and reshape it back to $(c_{out}, c_{in}, k, k)$. It is important to note that while the reshaped kernel differs from the orthogonal convolution described in BRO convolutional layer, it remains 1-Lipschitz bounded due to being non-overlapping (stride = kernel size $k$) (Tsuzuku et al., 2018).

- Dense: BRO or Cholesky-orthogonal (Hu et al., 2024) dense layers with width 2048 are appended to increase the network's depth and enhance the model capability.

- Head: The architecture concludes with an LLN (Last Layer Normalization) layer, an affine layer that outputs the prediction logits.

We can use the $W$, $L$, and $D$ to control the model size.

**LipConvNet Architecture** This architecture is utilized in orthogonal neural networks such as SOC and LOT. The fundamental architecture, LipConvNet, consists of five orthogonal convolutional blocks, each serving as a down-sampling layer. The MaxMin or householder (Singla et al., 2022) activation function is employed for activation, and the final layer is an affine layer such as LLN. Figure 7 provides an illustration of LipConvNet. To increase the network depth, dimension-preserving orthogonal convolutional blocks are added subsequent to each down-sampling block; thus, the depth remains a multiple of five. We use the notation LipConvNet-N to describe the depth, where N represents the number of layers. For example, LipConvNet-20 indicates a network with 20 layers, consisting of five down-sampling orthogonal layers and 15 dimension-preserving orthogonal layers.

### B.3 ARCHITECTURE AND RANK-n CONFIGURATION

As mentioned in Section 4.1, for BRO layers with dimension $d_{out} = d_{in} = m$, we explicitly set the unconstrained parameter $V$ to be of shape $m \times n$ with $m > n$ to avoid the degenerate case. For the BRONet-M backbone and dense layers, we set $n = m/4$ for CIFAR-10 and CIFAR-100 and $n = m/8$ for Tiny-ImageNet experiments. For the BRONet-L architecture, we use $n = m/2$ for for the BRO backbone and use Cholesky-orthogonal dense layers. For

LipConvNet, we set $n = m/8$ for all experiments. An ablation study on the effect of different choices of rank-n is presented in Appendix D.2.

### B.4 LA Hyper-parameters

Unless particularly specified, the LA loss hyper-parameters $T$, $\xi$, and $\beta$ are set to 0.75, $2\sqrt{2}$, and 5.0, respectively. The hyper-parameters are selected by an ablation experiments done on LipConvNet. Please see Appendix D.4 for the experiments.

### B.5 Table 1 Details

On CIFAR-10 and CIFAR-100, BRONet is configured with L12W512D8, and on Tiny-ImageNet, it is L6W512D4. Mainly following Hu et al. (2024), we use NAdam (Dozat, 2016) and the LookAhead Wrapper (Zhang et al., 2019) with an initial learning rate of $10^{-3}$, batch size of 256, and weight decay of $4 \times 10^{-5}$. The learning rate follows a cosine decay schedule with linear warm-up during the first 20 epochs, and the model is trained for a total of 800 epochs. We combine the LA loss with the EMMA (Hu et al., 2023) method to adjust non-ground-truth logit values for Lipschitz regularization on the stem layer. The target budget for EMMA is set to $\varepsilon = 108/255$ and offset for LA is set to $\xi = 2$. To report the results of LiResNet (Hu et al., 2024), we reproduce the results without diffusion data augmentation for fair comparison. All experimental results are the average of three runs. For other baselines, results are reported as found in the literature.

### B.6 Table 2 Details

In this table, we utilize diffusion-synthetic datasets from Hu et al. (2024); Wang et al. (2023) for CIFAR-10 and CIFAR-100, which contain 4 million and 1 million images, respectively. Following Hu et al. (2024), we employ a 1:3 ratio of real to synthetic images for each mini-batch, with a total batch size of 1024. We have removed weight decay, as we observed it does not contribute positively to performance with diffusion-synthetic datasets. All other settings remain consistent with those in Table 1.

### B.7 Table 3 Details

The settings are consistent with those in Table 2, where we use the default architecture of LiResNet (L12W512D8), LA loss, and diffusion data augmentation. We replace the convolutional backbone for each Lipschitz layer.

### B.8 Table 4 Details

Following the training configuration of Singla & Feizi (2021b), we adopt the SGD optimizer with an initial learning rate of 0.1, which is reduced by a factor of 0.1 at the 50-th and 150-th epochs, over a total of 200 epochs. Weight decay is set to $3 \times 10^{-4}$, and a batch size of 512 is used for the training process. The architecture is initialized with initial channel sizes of 32, 48, and 64 for different rows in the table. The LA loss is adopted for training.

## C Logit Annealing Loss Function

In this section, we delve into the details of the LA loss. Initially, we will prove Theorem 1, which illustrates the lower bound of the empirical margin loss risk. Next, we will visualize the LA loss and its gradient values. Additionally, we will discuss issues related to the CR term used in the SOC and LOT frameworks. Lastly, we will thoroughly explain the annealing mechanism.

### C.1 Empirical Margin Loss Risk

Here, we explain Theorem 1, which demonstrates how model capacity constrains the optimization of margin loss. The margin operation is defined as follows:

$$\mathcal{M}_f = f(x)_t - \max_{k \neq t} f(x)_k. \tag{19}$$

This operation is utilized to formulate margin loss, which is employed in various scenarios to enhance logit distance and predictive confidence. The margin loss can be effectively formulated using the *ramp loss* (Bartlett et al., 2017), which offers a analytic perspective on margin loss risk. Ramp loss provides a linear transition between full penalty and no penalty states. It is defined as follows:

$$\ell_{\tau,\mathrm{ramp}}(f, x, y) = \begin{cases} 0 & \text{if } f(x)_t - \max_{k \neq t} f(x)_k \geq \tau, \\ 1 & \text{if } f(x)_t - \max_{k \neq t} f(x)_k \leq 0, \\ 1 - \frac{f(x)_t - \max_{k \neq t} f(x)_k}{\tau} & \text{otherwise.} \end{cases}$$

We employ the margin operation and the ramp loss to define margin loss risk as follows:

$$\mathcal{R}_\tau(f) := \mathbb{E}(\ell_{\tau,\mathrm{ramp}}(\mathcal{M}(f(x), y))), \tag{20}$$

$$\hat{\mathcal{R}}_\tau(f) := \frac{1}{n} \sum_i \ell_{\tau,\mathrm{ramp}}(\mathcal{M}(f(x_i), y_i)), \tag{21}$$

where $\hat{\mathcal{R}}_\tau(f)$ denotes the corresponding empirical margin loss risk. According to Mohri et al. (2018), a risk bound exists for this loss:

**Lemma 4.** *(Mohri et al., 2018, Theorem 3.3) Given a neural network $f$, let $\tau$ denote the ramp loss. Let $\mathcal{F}$ represent the function class of $f$, and let $\mathfrak{R}_S(.)$ denote the Rademacher complexity. Assume that $S$ is a sample of size $n$. Then, with probability $1 - \delta$, we have:*

$$\mathcal{R}_\tau(f) \leq \hat{\mathcal{R}}_\tau(f) + 2\mathfrak{R}_S(\mathcal{F}_\tau) + 3\sqrt{\frac{\ln(1/\delta)}{2n}}. \tag{22}$$

Next, apply the following properties for the prediction error probability:

$$\mathcal{P}_e = \Pr\left[\arg\max_i f(x)_i \neq y\right] = \Pr\left[-\mathcal{M}(f(x), y) \geq 0\right] \tag{23}$$

$$= \mathbb{E}\mathbf{1}\left[\mathcal{M}(f(x), y) \leq 0\right] \tag{24}$$

$$\leq \mathbb{E}(\ell_{\tau,\mathrm{ramp}}(\mathcal{M}(f(x), y))) \tag{25}$$

$$= R_\tau(f), \tag{26}$$

where $\mathcal{P}_e$ is the prediction error probability. Assuming that the $\mathcal{P}_e$ is fixed but unknown, we can utilize Lemma 4 to prove Theorem 1:

$$\hat{\mathcal{R}}_\tau(f) \geq \mathcal{P}_e - 2\mathfrak{R}_S(\mathcal{F}_\tau) - 3\sqrt{\frac{\ln(1/\delta)}{2n}}. \tag{27}$$

This illustrates that the lower bound for the margin loss risk is constrained by model complexity.

Next, we illustrate and prove the relationship between margin loss risk and the CR loss risk. Let the empirical CR loss risk be defined as follows:

$$\hat{\mathcal{R}}_{CR}(f) := \frac{1}{n} \sum_i -\gamma \max(\mathcal{M}(f(x_i), y_i), 0). \tag{28}$$

**Proposition 4.** *Let $\hat{\mathcal{R}}_{CR}(f)$ and $\hat{\mathcal{R}}_\tau(f)$ are the CR loss risk and margin loss risk, respectively. Assume that $\tau = \sup_i \mathcal{M}_f(x_i)$ and $\gamma = 1/\tau$. Then, $\hat{\mathcal{R}}_{CR}(f)$ is $\hat{\mathcal{R}}_\tau(f)$ decreased by one unit:*

$$\hat{\mathcal{R}}_{CR}(f) = \hat{\mathcal{R}}_\tau(f) - 1. \tag{29}$$

*Proof.* (Proof for Proposition 4) Consider two cases based on the value of $\mathcal{M}(x)$:

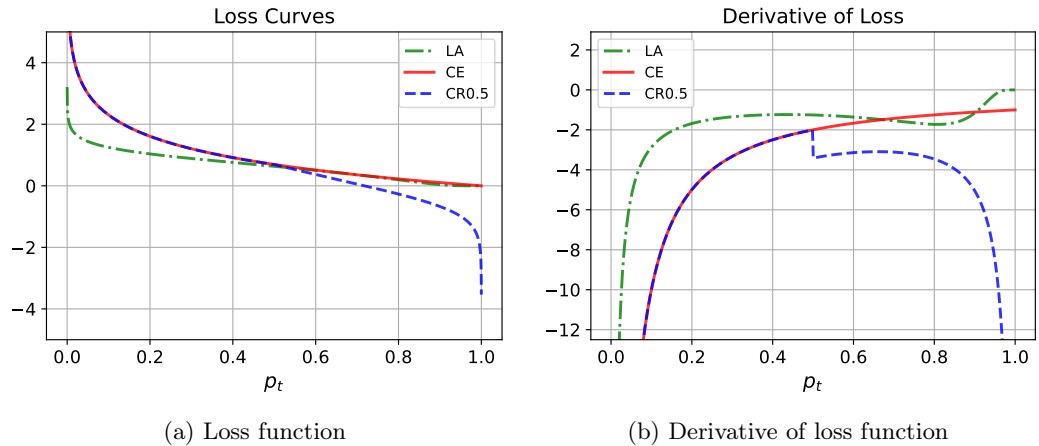

(a) Loss function

(b) Derivative of loss function

Figure 9: Comparison of three loss functions. The x-axis is $p_t$. This figure displays curves representing the behavior of the proposed LA loss, contrasted with cross-entropy loss and the Certificate Regularization (CR) term. We observe the discontinuous gradient of the CR term. Additionally, the gradient of the CR term tends to infinity as $p_t$ approaches one, leading to gradient domination and subsequently hindering the optimization of other data points. In contrast, the proposed LA loss employs a different strategy, where the gradient value anneals as nears one. This prevents overfitting and more effectively utilizes model capacity to enhance learning across all data points.

- When $\mathcal{M}(x) \leq 0$: the CR loss is always zero and the ramp loss is always one. Thus, the distance between $\hat{\mathcal{R}}_{CR}(f)$ and $\hat{\mathcal{R}}_\tau(f)$ is one.

- When $\mathcal{M}(x) > 0$: The distance between the ramp loss and CR loss is:

$$\ell_{\tau,\mathrm{ramp}}(\mathcal{M}(f(x_i), y_i)) + \gamma \max(\mathcal{M}(f(x_i), y_i), 0) = 1 - \frac{\mathcal{M}(x_i)}{\tau} + \gamma \mathcal{M}(x_i)$$

$$= 1 + (\gamma - \frac{1}{\tau})\mathcal{M}(x_i). \qquad (30)$$

Therefore, the empirical CR loss risk can be rewritten as:

$$\hat{\mathcal{R}}_{CR}(f) = \hat{\mathcal{R}}_\tau(f) - 1 - (\gamma - \frac{1}{\tau})M_+, \text{ where} \qquad (31)$$

$$M_+ = \sum\nolimits_{x_i \in \{x_i | \mathcal{M}(x_i) > 0\}} \mathcal{M}(x_i). \qquad (32)$$

This equation simplifies to the one stated in Proposition 4 if $\gamma = 1/\tau$. $\qquad \square$

Conclusively, we demonstrate that the CR loss risk has a lower bound as follows:

$$\hat{\mathcal{R}}_{CR}(f) \geq \mathcal{P}_e - 2\mathfrak{R}_S(\mathcal{F}_\tau) - 3\sqrt{\frac{\ln(1/\delta)}{2n}} - 1. \qquad (33)$$

When the complexity is limited, CR loss risk exhibits a great lower bound. This indicates that we cannot indefinitely minimize the CR loss risk. Thus, enlarging margins using the CR term is less beneficial beyond a certain point.

### C.2 CR Issues

Recall that CE loss with CR term is formulated as: $\mathcal{L}_{\mathrm{CE}} - \gamma \max(\mathcal{M}_f(x), 0)$, where $\mathcal{M}_f(x) = f(x)_t - \max_{k \neq t} f(x)_k$ is the logit margin between the ground-truth class $t$ and the runner-up class. We compare LA loss, CE loss, and the CE+CR loss with $\gamma = 0.5$. Figure 9 illustrates

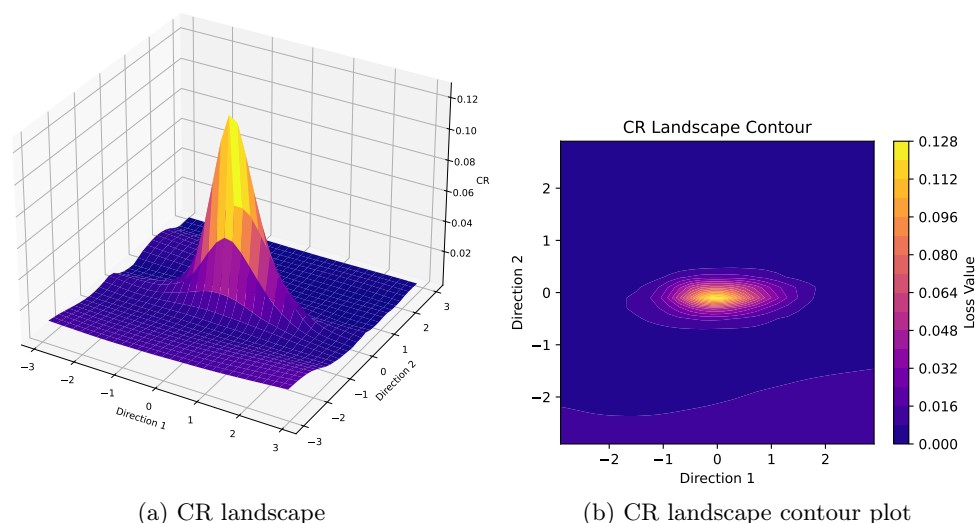

(a) CR landscape          (b) CR landscape contour plot

Figure 10: CR Loss Landscape Analysis. This figure illustrates the loss landscape to investigate the effects of the CR term. Notably, the CR term can suddenly become "activated" or "deactivated," which is vividly depicted in the landscape transitions. These abrupt changes contribute to unstable optimization during training, potentially affecting the convergence and reliability of the model. Understanding this behavior is crucial for improving the training process of Lipschitz neural networks. Regarding the direction of loss landscape, we follow the setting in Engstrom et al. (2018) and Chen et al. (2020). We visualize the loss landscape function $z = \mathcal{L}_{CR}(x, w + \omega_1 d_1 + \omega_2 d_2)$, where $d_1 = sign(\nabla_w \mathcal{L}_{CR})$, $d_2 \sim Rademacher(0.5)$, and $\omega$ is the grid.

the loss values and their gradient values with respect to $p_t$, where $p_t$ represents the softmax result of the target logit. When using CR term as the regularization for training Lipschitz models, we summarize the following issues:

(1). Discontinuous loss gradient: the gradient value of CR term at $p_t = 0.5$ is discontinuous This discontinuity leads to unstable optimization processes, as shown in Figure 9. This indicates that, during training, the CR loss term may be "activated" or "deactivated." This phenomenon can be further explored through the loss landscape. Figure 10 displays the CR loss landscape for the CR term, where it can be seen that the CR term is activated suddenly. The transition is notably sharp.

(2). Gradient domination: as $p_t$ approaches one, the gradient value escalates towards negative infinity. This would temper the optimization of the other data points in the same batch.

(3). Imbalance issue: our observations indicate that the model tends to trade clean accuracy for increased margin, suggesting a possible imbalance in performance metrics.

Therefore, instead of using the CR term to train Lipschitz neural networks, we design the LA loss to help Lipschitz models learn better margin values.

## C.3 ANNEALING MECHANISM

We can observe the annealing mechanism in the right subplot of Figure 9. The green curve is the gradient value of the LA loss. We can observe that the gradient value is gradually annealed to zero as the $p_t$ value approaches one. This mechanism limits the optimization of the large-margin data points. As mentioned previously, Lipschitz neural networks have limited capacity, so we cannot maximize the margin indefinitely. Since further enlarging

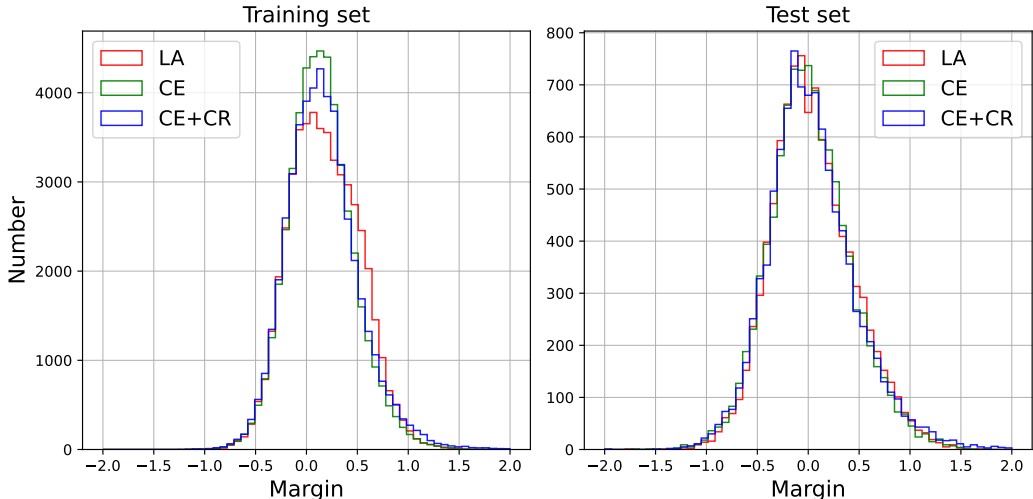

Figure 11: Histogram of margin distribution. The left histogram represents margin distribution obtained from the training set, while the right histogram shows margin distribution from the test set. The x-axis represents the margin values. These visualizations demonstrate that the LA loss helps the model learn better margins.

Table 8: The clean, certified, and empirical robust accuracy of BRONet-M on CIFAR-10, CIFAR-100, and Tiny-ImageNet.

| Datasets | Clean Acc. | Certified / AutoAttack ($\varepsilon$) | | |
|---|---|---|---|---|
| | | $\frac{36}{255}$ | $\frac{72}{255}$ | $\frac{108}{255}$ |
| CIFAR10 | 81.1 | 69.9 / 76.1 | 55.3 / 69.7 | 40.4 / 62.6 |
| CIFAR100 | 54.3 | 40.0 / 47.3 | 28.7 / 41.0 | 19.4 / 35.5 |
| Tiny-ImageNet | 41.0 | 29.2 / 36.3 | 19.7 / 31.7 | 12.3 / 27.5 |

the margin for data points with sufficiently large margin is less beneficial, we employ the annealing mechanism to allocate the limited capacity for the other data points.

In addition, we delve deeper into the annealing mechanism of the proposed LA loss function. As illustrated in Figure 11, we train three different models using three loss functions, and we plot the histogram of their margin distribution. The red curve represents the proposed LA loss. Compared to CE loss, the proposed LA loss has more data points with margins between 0.4 and 0.8. This indicates that the annealing mechanism successfully improves the small-margin data points to appropriate margin 0.4 and 0.8.

Additionally, as the left subplot in Figure 11 illustrates, the margin exhibits an upper bound; no data points exceed a value of 2.0, even when a larger $\gamma$ is used in the CR term. This observation coincides with our theoretical analysis, confirming that the Lipschitz models cannot learn large margins due to its limited capacity.

## D  ADDITIONAL EXPERIMENTS

In this section, we present additional experiments and ablation studies.

### D.1  EMPIRICAL ROBUSTNESS

In addition to certified robustness, we can validate the empirical robustness of the proposed method. This further supports our robustness certificate. Theoretically, certified robust accuracy is the lower bound for the worst-case accuracy, while empirical robust accuracy is the upper bound for the worst-case accuracy. Thus, empirical robust accuracy must be

Table 9: We compare the clean accuracy, certified accuracy, and training time for different choices of $n$ for the unconstrained parameter $V$ on CIFAR-100 with BRONet L6W256D4 and L6W512D4. Time is calculated in minutes per training epoch.

| n | L6W256D4 | | | | | L6W512D4 | | | | |
|---|---|---|---|---|---|---|---|---|---|---|
| | Clean | $\frac{36}{255}$ | $\frac{72}{255}$ | $\frac{108}{255}$ | Time | Clean | $\frac{36}{255}$ | $\frac{72}{255}$ | $\frac{108}{255}$ | Time |
| $m/8$ | 51.6 | 39.2 | 28.3 | 19.5 | 0.66 | 52.8 | 40.2 | 28.6 | 20.3 | 1.57 |
| $m/4$ | 52.8 | 39.5 | 27.9 | 19.7 | 0.73 | 54.0 | 40.2 | 28.3 | 19.3 | 1.92 |
| $m/2$ | 53.4 | 39.0 | 27.3 | 18.5 | 0.94 | 54.1 | 39.7 | 27.7 | 18.6 | 2.82 |
| $3m/4$ | 52.7 | 39.5 | 28.0 | 19.2 | 1.27 | 53.5 | 39.8 | 27.9 | 18.9 | 3.75 |

Table 10: The improvement of LA loss with BRONet-M on different datasets.

| Datasets | Loss | Clean | $\frac{36}{255}$ | $\frac{72}{255}$ | $\frac{108}{255}$ |
|---|---|---|---|---|---|
| CIFAR10 | $CE_{EMMA}$ | 81.8 | 68.9 | 53.6 | 38.3 |
| | $\mathbf{LA_{EMMA}}$ | 81.2 | 69.7 | 55.6 | 40.7 |
| CIFAR100 | $CE_{EMMA}$ | 54.7 | 38.9 | 26.3 | 16.7 |
| | $\mathbf{LA_{EMMA}}$ | 54.1 | 40.1 | 28.5 | 19.6 |
| Tiny-ImageNet | $CE_{EMMA}$ | 40.5 | 26.9 | 17.1 | 10.1 |
| | $\mathbf{LA_{EMMA}}$ | 41.2 | 29.0 | 19.0 | 12.1 |

Table 11: The improvement of LA loss with LipConvNet on different datasets.

| Datasets | Loss | Clean | $\frac{36}{255}$ | $\frac{72}{255}$ | $\frac{108}{255}$ |
|---|---|---|---|---|---|
| CIFAR10 | CE | 77.5 | 62.1 | 44.8 | 29.2 |
| | **LA** | 76.9 | 63.4 | 47.2 | 32.6 |
| CIFAR100 | CE | 48.5 | 34.1 | 22.6 | 14.4 |
| | **LA** | 48.6 | 35.4 | 24.5 | 16.1 |
| Tiny-ImageNet | CE | 38.0 | 26.3 | 17.0 | 10.3 |
| | **LA** | 39.4 | 28.1 | 18.2 | 11.6 |

greater than certified robust accuracy. We employ AutoAttack (Croce & Hein, 2020) to assess empirical robustness. The certified and empirical robust accuracy for different attack budgets are illustrated in Table 8. We observe that all empirical robust accuracy values for each budget are indeed higher than their corresponding certified accuracy. This indicates that the certification is correct under the AutoAttack test. Additionally, Table 8 shows that the proposed method achieves strong empirical robustness without any adversarial training techniques.

## D.2 BRO Rank-n Ablation Experiments

As mentioned earlier, we can control the rank of $V$ to construct the orthogonal weight matrix. In this paper, the matrix $V$ is of low rank. Considering the internal term $V(V^T V)^{-1}V^T$ in our method's parameterization, the concept is similar to that of LoRA (Hu et al., 2021). We further investigate the effect of different $n$ values of $V$. For the unconstrained $m \times n$ parameter $V$ in the backbone and dense blocks of BRONet, we conduct experiments using different $n$ values. The clean and certified accuracy, as well as training time, on CIFAR-100 are presented in Table 9. Different $n$ values result in slightly different performance. Therefore, we choose $n = m/4$ for all CIFAR-10/CIFAR-100 experiments on BRONet-M, and $n = m/2$ for BRONet-L. For TinyImageNet, considering our computational resources, we choose $n = m/8$ to save memory, as the $n$ values help control memory usage.

## D.3 LA Loss Ablation Experiments

Table 10 presents an ablation study on the effectiveness of the proposed LA loss function. We train BRONet-M using both the original CE-based EMMA loss, as described in Hu et al. (2023), and the newly proposed LA-based EMMA loss. By switching from CE to LA, we achieve an improvement in certified accuracy for all $\ell_2$ perturbations by approximately 1.94% on average while maintaining the same level of clean accuracy.

Moreover, we verify the LA loss on LipConvNet constructed using BRO, LOT, or SOC. Table 12 illustrates the improvement achieved by replacing the CE+CR loss, which is initially recommended for training LipConvNet. The results suggest that using the LA loss improves the performance of LipConvNet constructed with all three orthogonal layers on both CIFAR-100 and Tiny-ImageNet.

Table 12: Comparison of clean and certified accuracy, training and inference time (seconds/epoch), and number of parameters with different orthogonal layers in LipConvNet-10. Instances marked with a dash (-) indicate out of memory during training. In the Time column, we show the training time, and the inference time is in brackets.

| Init. Width | Methods | CIFAR-100 | | | | | Tiny-ImageNet | | | | |
|---|---|---|---|---|---|---|---|---|---|---|---|
| | | Clean | $\frac{36}{255}$ | $\frac{72}{255}$ | $\frac{108}{255}$ | Time | Clean | $\frac{36}{255}$ | $\frac{72}{255}$ | $\frac{108}{255}$ | Time |
| 32 | SOC + CR | 48.1 | 34.3 | 23.5 | 15.6 | 19.2 | 37.4 | 26.2 | 17.3 | 11.2 | 107.7 |
| | LA | 47.5 | 34.7 | 24.0 | 15.9 | (5.3) | 38.0 | 26.5 | 17.7 | 11.3 | (11.1) |
| | LOT + CR | 48.8 | 34.8 | 23.6 | 15.8 | 52.7 | 38.7 | 26.8 | 17.4 | 11.3 | 291.5 |
| | LA | 49.1 | 35.5 | 24.4 | 16.3 | (1.4) | 40.2 | 27.9 | 18.7 | 11.8 | (7.3) |
| | **BRO** + CR | 48.4 | 34.7 | 23.6 | 15.4 | 17.3 | 38.5 | 27.1 | 17.8 | 11.7 | 98.6 |
| | LA | 48.6 | 35.4 | 24.5 | 16.1 | (0.9) | 39.4 | 28.1 | 18.2 | 11.6 | (4.6) |
| 48 | SOC + CR | 48.4 | 34.9 | 23.7 | 15.9 | 35.4 | 38.2 | 26.6 | 17.3 | 11.0 | 199.3 |
| | LA | 48.2 | 34.9 | 24.4 | 16.2 | (8.7) | 38.9 | 27.1 | 17.6 | 11.2 | (20.3) |
| | LOT + CR | 49.3 | 35.3 | 24.2 | 16.3 | 143.0 | - | - | - | - | - |
| | LA | 49.4 | 35.8 | 24.8 | 16.3 | (3.0) | - | - | - | - | |
| | **BRO** + CR | 49.4 | 35.7 | 24.5 | 16.3 | 35.2 | 38.9 | 27.2 | 18.0 | 11.6 | 196.9 |
| | LA | 49.4 | 36.2 | 24.9 | 16.7 | (1.1) | 40.0 | 28.1 | 18.9 | 12.3 | (4.8) |
| 64 | SOC + CR | 48.4 | 34.8 | 24.1 | 16.0 | 53.1 | 38.6 | 26.9 | 17.3 | 11.0 | 305.1 |
| | LA | 48.5 | 35.5 | 24.4 | 16.3 | (12.4) | 39.3 | 27.3 | 17.6 | 11.2 | (32.5) |
| | LOT + CR | 49.4 | 35.4 | 24.4 | 16.3 | 301.8 | - | - | - | - | - |
| | LA | 49.6 | 36.1 | 24.7 | 16.2 | (5.8) | - | - | - | - | |
| | **BRO** + CR | 49.7 | 35.6 | 24.5 | 16.4 | 64.4 | 39.6 | 27.9 | 18.2 | 11.9 | 355.3 |
| | LA | 49.7 | 36.7 | 25.2 | 16.8 | (1.6) | 40.7 | 28.4 | 19.2 | 12.5 | (4.9) |

Table 13: Experimental results for LipConvNet-10 on CIFAR-100 for different values of $\beta$ in the LA loss.

| $\beta$ value | Clean | $\frac{36}{255}$ | $\frac{72}{255}$ | $\frac{108}{255}$ |
|---|---|---|---|---|
| 1 | 48.63 | 35.48 | 24.36 | 17.19 |
| 3 | 48.57 | 35.68 | 24.78 | 16.66 |
| 5 | 49.09 | 35.58 | 24.46 | 16.38 |
| 7 | 49.02 | 35.72 | 24.34 | 16.05 |

We also compare LA to CE on LipConvNet. Table 11 shows the results for LipConvNet constructed with BRO. Our results show that the LA loss encourages a moderate margin without compromising clean accuracy. Notably, the LA loss is more effective on larger-scale datasets, suggesting that the LA loss effectively addresses the challenge of models with limited Rademacher complexity.

## D.4 LA Loss Hyper-parameters Experiments

There are three tunable parameters in LA loss: temperature $T$, offset $\xi$, and annealing factor $\beta$. The first two parameters control the trade-off between accuracy and robustness, while the last one determines the strength of the annealing mechanism. For the temperature and offset, we slightly adjust the values used in Prach & Lampert (2022) to find a better trade-off position, given the differences between their network settings and ours. Additionally, we present the results of LA loss with different $\beta$ values for CIFAR-100 on LipConvNet in Table 13.

## D.5 LipConvNet Ablation Experiments

More detailed comparison stem from Table 4 are provided in Table 12, demonstrating the efficacy of LA loss across different model architectures and orthogonal layers. Following the

Table 14: The experiments conducted with varying initial widths and model depths using the CIFAR-100 and Tiny-ImageNet datasets. The model employed is LipConvNet.

| Depth | Init. Width | CIFAR-100 | | | | Tiny-ImageNet | | | |
|---|---|---|---|---|---|---|---|---|---|
| | | Clean | $\frac{36}{255}$ | $\frac{72}{255}$ | $\frac{108}{255}$ | Clean | $\frac{36}{255}$ | $\frac{72}{255}$ | $\frac{108}{255}$ |
| 5 | 32 | 49.04 | 35.06 | 24.19 | 16.06 | 39.28 | 27.47 | 18.23 | 11.47 |
| | 48 | 49.60 | 35.80 | 24.63 | 16.20 | 40.12 | 27.79 | 18.36 | 11.92 |
| | 64 | 49.97 | 36.21 | 24.92 | 16.45 | 40.82 | 28.26 | 18.76 | 12.31 |
| 10 | 32 | 48.62 | 35.36 | 24.48 | 16.11 | 39.37 | 28.06 | 18.16 | 11.58 |
| | 48 | 49.39 | 36.19 | 24.86 | 16.68 | 39.98 | 28.12 | 18.86 | 12.17 |
| | 64 | 49.74 | 36.70 | 25.24 | 16.80 | 40.66 | 28.36 | 19.24 | 12.48 |
| 15 | 32 | 48.59 | 35.51 | 24.42 | 16.28 | 39.20 | 27.66 | 18.08 | 11.84 |
| | 48 | 49.37 | 36.50 | 24.93 | 16.81 | 39.87 | 27.96 | 18.49 | 12.11 |
| | 64 | 49.91 | 36.57 | 25.26 | 16.81 | 40.38 | 28.73 | 18.78 | 12.52 |
| 20 | 32 | 48.62 | 35.68 | 24.66 | 16.57 | 38.74 | 27.23 | 17.75 | 11.67 |
| | 48 | 49.26 | 36.09 | 24.91 | 16.62 | 39.63 | 27.88 | 18.49 | 12.07 |
| | 64 | 49.60 | 36.47 | 25.24 | 17.09 | 39.77 | 28.03 | 18.53 | 12.17 |

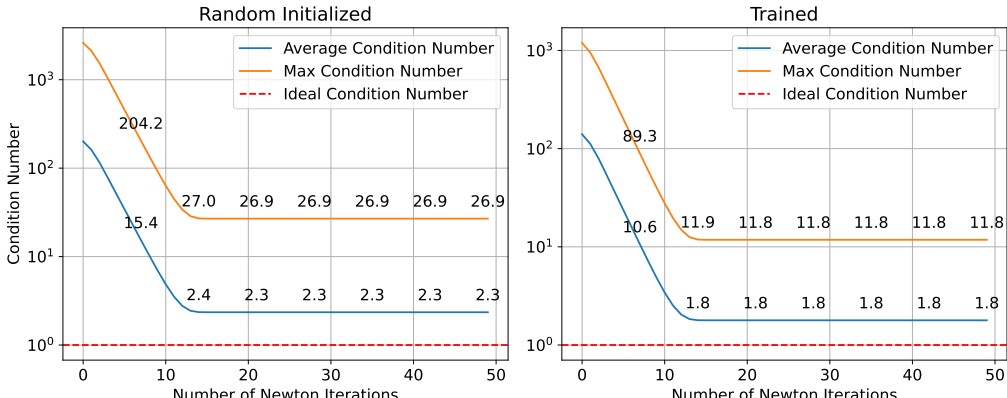

Figure 12: Plots of condition number of parameterized matrix in Fourier domain. The left plot shows the condition number with randomly initialized parameters, whereas the right plot shows the condition number with trained parameters.

same configuration as in Table 4, we further investigate the construction of LipConvNet by conducting experiments with varing initial channels and model depths, as detailed in Table 14.

### D.6 STABILITY OF LOT PARAMETERIZATION

During the construction of the LOT layer, we empirically observed that replacing the identity initialization with the common Kaiming initialization for dimension-preserving layers causes the Newton method to converge to a non-orthogonal matrix. We check orthogonality by computing the condition number of the parameterized matrix of LOT in the Fourier domain. For an orthogonal layer, the condition number should be close to one. However, even after five times the iterations suggested by the authors, the result for LOT does not converge to one. Figure 12 illustrates that, even with 50 iterations, the condition number of LOT does not converge to one. The orange curve represents the case with Kaiming randomly initialized parameters, while the blue curve curve corresponds to the case after a few training epochs. Both exhibit a significant gap compared to the ideal case, indicating that LOT may produce a non-orthogonal layer.

## E  LIMITATIONS

★ This is a newly added section. ★

While our proposed methods have demonstrated improvements across several metrics, the results for large perturbations, such as $\varepsilon = 108/255$, are less consistent. Additionally, the proposed LA loss requires extra hyperparameter tuning. In our experiments, the parameters were chosen based on LipConvNets trained on CIFAR-100 without diffusion-synthetic augmentation (Appendix B.4), which may not fully align with different models and datasets. Furthermore, our methods are specifically designed for $\ell_2$ certified robustness, and certifying against attacks like $\ell_\infty$-norm introduces additional looseness. Lastly, although BRO addresses some limitations of orthogonal layers, training certifiably robust models on large datasets, such as ImageNet, remains computationally expensive and beyond our current resources.

