# OpenReview forum: "Enhancing Certified Robustness via Block Reflector Orthogonal Layers"
_ICLR.cc/2025/Conference — Submitted to ICLR 2025_

### Official Review · Reviewer_9JSB · 2024-10-23

**Soundness:** 2
**Presentation:** 3
**Contribution:** 2
**Rating:** 6
**Confidence:** 3

**Summary:**

This paper presents the block reflector orthogonal (BRO) layer, which enhances the construction of Lipschitz neural networks. By employing low-rank parameterization and circumventing iterative approximations, the proposed approach achieves notable gains in both memory and time efficiency. Comprehensive evaluations against existing orthogonal layers reveal its superior robustness, underscoring its potential in advancing neural network performance.

**Strengths:**

1. The BRO method effectively utilizes low-rank parameterization to construct orthogonal layers, resulting in significant improvements in both computational time and memory efficiency, which are critical for scaling neural networks.
2. The paper provides a thorough comparison with state-of-the-art techniques, presenting results that are well-structured and articulate, allowing readers to easily grasp the contributions and effectiveness of the proposed method.
3. The paper is well-structured and logically organized. The presentation of the results is clear and systematic.

**Weaknesses:**

1. The evaluation results in Table 1 indicate a degradation in the performance of the proposed BRONet with increasing perturbation budgets. A discussion on this inconsistency would enhance the paper's credibility and provide insight into the limitations of the model.
2. Results in Table 3 show only marginal improvements over existing methods, raising concerns about the significance of these gains. Including standard deviations or confidence intervals would clarify the statistical significance of the results and help determine whether observed improvements are due to random variance in training.
3. The experiments are conducted solely on CIFAR and Tiny-ImageNet datasets, which, while widely used, may not fully demonstrate the method's robustness and scalability. Including additional datasets, particularly larger or more complex ones (e.g., ImageNet or real-world benchmarks), would provide a more comprehensive evaluation of the method's efficacy.

**Questions:**

1. What factors contribute to the inconsistent performance gains observed in the proposed BRONet as the perturbation budget increases?
2. Why were standard deviations or confidence intervals not included in the reported results? Given the marginal improvements, understanding the variability of the outcomes would be particularly valuable.
3. Has the BRO method been evaluated on more complex datasets, such as full ImageNet, or applied to tasks beyond classification, to assess its scalability and generalization?

---

> ### Author Response · Authors · 2024-11-18
> **Response for Reviewer 9JSB**
>
> **Q: Inconsistent performance gains in BRONet as the perturbation budget increases?**
>
> Thank you for your valuable suggestions and response. Ideally, we aim to build a model with state-of-the-art certified robustness with the constraint that it must maintain high clean accuracy. In this work, we focus on developing models that can achieve better margins for most data points. However, improving robustness to larger perturbations without compromising clean accuracy or robustness for smaller perturbations remains a challenge for future work.
>
>
> **Q: Has the BRO method been evaluated on larger datasets or tasks beyond classification to assess its scalability and generalization?**
>
> Thank you for your insightful question. While BRO has not yet been evaluated on more complex datasets or applied to tasks beyond classification, exploring its potential in these areas is an intriguing direction for future research. Given that certified robustness have been utilized in various computer vision tasks, BRO presents a strong candidate for providing competitive certified robustness in these domains. We believe that such extensions may facilitate the practical application of our method and hope that our results lay the foundation for future research on this topic.
>
> **Q: Regarding reporting statistics**
>
> Thank you for your suggestions. Following prior works [1][2], we chose to report the average of three runs. Although BRO mitigates some of the resource requirements compared to other orthogonal layers, training a certified robust model remains expensive and can take several days given our computational resources, even on a smaller-scale dataset such as CIFAR-10.
>
> Overall, we observed that the standard deviations are sufficiently small compared to the improvements in clean and certified accuracy at $\varepsilon = 36/255$ (+0.6, +0.8). For reference, the standard deviations for the CIFAR-10 experiments in Table 1 is as follows: 0.035, 0.020, 0.359, and 0.191 for clean and certified accuracies across budgets $\varepsilon = 36/255, 72/255, 108/255$, respectively.
>
> In future revisions, we plan to conduct additional runs and report the confidence intervals to strengthen the results further.
>
> [1] Hu et al. "A Recipe for Improved Certifiable Robustness." International Conference on Learning Representations (ICLR), 2024.
>
> [2] Araujo et al. "A Unified Algebraic Perspective on Lipschitz Neural Networks." International Conference on Learning Representations (ICLR), 2023.

---

> > ### Comment · Reviewer_9JSB · 2024-11-24
> >
> > Thank you for the authors' response. However, some concerns remain:
> >
> > 1. Regarding the inconsistency of the results, my comment is not about robustness to larger perturbations. Instead, I am referring to the inconsistent improvements of the proposed method itself, as illustrated in Table 1 and highlighted in Weakness 1.
> >
> > 2.  In my original comments, I highlighted the marginal improvements reported in Table 3. While the authors have provided standard deviations for Table 1, applying the same analysis to Table 3 would reveal that the reported improvements are relatively minor.
> >
> > Additionally, I share concerns raised by Reviewer 1mnT regarding the rigorous proof of the orthogonality of BRO convolution. To substantiate the claim of orthogonal layers, a rigorous proof of the orthogonality is essential. While the additional results provided are appreciated, a more rigorous proof addressing Reviewer 1mnT's concerns is critical to my recommendation for acceptance.

---

> > > ### Author Response · Authors · 2024-11-25
> > >
> > > **Regarding the first concern**
> > >
> > > Thank you for your feedback. Regarding the first concern, we understand your question and apologize for any ambiguity in the previous response, where we would like to emphasize that metrics for smaller perturbations should have a higher priority.
> > > As demonstrated in Table 1, BRO shows less consistent improvements for large perturbations. We have now included this along with a discussion on the limitation of our work in Appendix E.
> > >
> > > **Regarding Table 3**
> > >
> > > We apologize for the oversight. The analysis for CIFAR10+EDM (Table 3) is as follows:
> > > For clean and certified accuracies at budgets $\varepsilon = 36/255$ and $72/255$, the standard deviations are relatively small compared to the improvements (+0.5, +0.2, +0.4) over the baseline Lip-reg. For reference, the standard deviations for clean and certified accuracies across budgets $\varepsilon = 36/255$, $72/255$, and $108/255$ are 0.020, 0.026, 0.010, and 0.109, respectively.
> > >
> > >
> > > **Orthogonality**
> > >
> > > In response to Reviewer 1mnT's concerns regarding the orthogonality of the proposed BRO convolution, we have revised the manuscript to address this issue. Proposition 2 has been updated to clearly illustrate and emphasize the orthogonality of BRO convolution. Additionally, we have included a detailed proof of BRO's orthogonality in Appendix A.2. This derivation follows the structure established in prior work [1], providing a solid foundation to validate our claims.
> > >
> > > [1] Trockman et al. "Orthogonalizing Convolutional Layers with the Cayley Transform." International Conference on Learning Representations(ICLR), 2020.

---

### Official Review · Reviewer_1mnT · 2024-10-28

**Soundness:** 1
**Presentation:** 1
**Contribution:** 1
**Rating:** 3
**Confidence:** 4

**Summary:**

This paper proposes a novel 1-Lipschitz layer based on Block-reflector matrices (BRO convolution), which are orthogonal. Additionally, authors propose a Logit Annealing (LA) loss to optimize during training in order to favor certified accuracy. Their proposed recipe consistently improves certified accuracy in CIFAR10/100 and TinyImageNet and is shown to be more efficient than previously proposed orthogonal layers.

**Strengths:**

The strengths of this work are mainly on the experimental side. Assuming the theoretical results hold:

- More efficient and scalable models than the previous art due to exact 1-Lipschitz layers without the need of iterative approximations.
- Improved certified accuracy. Authors demonstrate that both their BRO convolutional layer and LA loss help improving the certified accuracy.

**Weaknesses:**

The weaknesses of the paper appear on the theoretical and implementation sides. In general, the paper is very hard to read and has many errors in key parts of the paper. This questions the validity of their 1-Lipschitz claims for the BRO convolution.

- **Many typos and unclear notation:**
	- Everywhere: Please use $c_{\text{in}}$ and not $c_{in}$ when subscripting or upperscripting with words.
	- Algorithm 1, line 1: I assume $c_{\text{in}}$ and $c_{\text{in}}’$ are the number of input channels for the current and next layers respectively. Then, why do you need $c_{\text{out}}$? It is very confusing. I would just remove $c_{\text{in}}’$ and use $c_{\text{out}}$ instead.
	- Algorithm 1, line 4: $\tilde{V}:= \text{FFT}(V)$ should be $\tilde{V}:= \text{FFT}(V^{\text{pad}})$.
	- Lines 205 and 214: Is $\circledast$ the convolution operator?
	- Proposition 2: If $J \in \mathbb{C}^{m \times m}$, then $J^* \in \mathbb{C}^{m \times m}$ as well. How can you even perform $J\tilde{V}J^{*}$ if $\tilde{V} \in \mathbb{C}^{m\times n}$? Does it have to be that $m=n$?
	- Proposition 2, equation 14: If assuming $n=m$, the proof is right, but equation 14 should be: $(\tilde{V}^* \tilde{V})^{-1} = J(J\tilde{V}^*\tilde{V}J^*)^{-1}J^*$

- **Unclear if BRO convolutions are orthogonal:**

From Proposition 1, it’s clear that in the dense case, the BRO layer is orthogonal. However, in the convolutional case there are many unclear aspects.

In lines 205 and 214, authors state that they construct their BRO convolution based on constructing the kernel:
	$$
		W_{\text{Conv}} = I - 2V \circledast (V^{\top} \circledast V)^{-1} \circledast V^{\top}
	$$
Authors argue that the circular convolution with this kernel is orthogonal, but do not provide any proofs. Then, given $\tilde{V} = \text{FFT}(V)$ authors conclude $\tilde{W} = \text{FFT}(W_{\text{Conv}}) = I - 2\tilde{V}(\tilde{V}^*\tilde{V})^{-1}\tilde{V}^*$. It’s unclear how authors arrive to this expression as it would need to assume that: (i) $n=m$ to be able to do element wise products (which authors do not mark with $\cdot$, leading to confusion) (ii) $\text{FFT}(I)=I$ (iii) $\text{FFT}((V^{\top} \circledast V)^{-1}) = \text{FFT}((V^{\top} \circledast V))^{-1}$, which are false in general.

Moreover, it is not clear what Proposition 2 implies regarding orthogonality, or how the case when $m=n$ is handled (lines 237-241 are very vague). It would be nice to include an explicit proof that the norm of the output of algorithm 1 is the same as the input. Exactness of the 1-Lipschitz result is important for ensuring the experimental results about the certified accuracy are valid. So, I believe more rigour should be put into proving the results claimed in this paper.

- **Unclear motivation to use LA and differences with CR:**

Authors start motivating the need of their LA loss with a very convoluted argument about the minimal CR loss being constrained by the model complexity. Then, they simply present their loss without relating it with the theory they developed for the CR loss. I believe the analysis doesn’t add anything if it is not performed for the LA loss. While I see that LA is less strict I don’t see how LA solves the issue in Theorem 1.

It would be more beneficial to fully state the CR loss and compare it with LA. Right now, only the reference is provided. I also think the results in Figure 8 and section C.2 are more interesting than the current write-up of the main paper.

All in all, I believe authors should be more careful about their theoretical derivations and be very clear about why Algorithm 1 results in orthogonal convolutions. Without this being clear, the results are meaningless and I cannot propose the paper for acceptance.

**Questions:**

Please see the weaknesses section for questions about the theory, notation and typos.

- Can authors implement the convolution without the FFT? Just creating $W_{\text{Conv}}$ and applying the standard Conv2D function from torch. Are the results the same as in Algorithm 1? Your outputs should be the same and its a good way to debug if anything is wrong.
- Did authors try LiResNet + BRO only for table 2? That is, only changing the backbone without changing the loss.
- The BRO layer is for sure orthogonal in the case of fully connected layers. Have authors tried comparing the different layers in the fully connected case? I.e. training fully connected classifiers in a small dataset (Cifar10 / MNIST). I don’t believe this experiment is super interesting but is the only case where orthogonality is clear at the moment.

---

> ### Author Response · Authors · 2024-11-18
> **Response for Reviewer 1mnT**
>
> **Q: The orthogonality of BRO convolution**
>
> We apologize for any statements and typos that may have obscured the orthogonality of BRO convolution, we have addressed them in the revised version.
>
> First, we defined $W_{\text{Conv}}$ as
> $$
> I_{\text{Conv}} - 2V \circledast (V^{\top} \circledast V)^{-1} \circledast V^{\top},
> $$
> where $I_{\text{Conv}}$ represents the convolution identity kernel.
>
> Our approach builds upon prior works, such as Cayley and LOT, based on the key idea that *multi-channel convolution in the Fourier domain can be reduced to a batch of matrix-vector products. By ensuring that each of these matrices is orthogonal, the convolution operation itself becomes orthogonal.*
>
> The following pipeline serves as a constructive proof for the orthogonality of BRO convolution:
>
> 1. **Fourier Transform:** Apply the Fourier transform to both the parameters and the input.
> 2. **BRO Matrix Multiplications:** Perform BRO matrix multiplications in the Fourier domain.
> 3. **Inverse Fourier Transform:** Apply the inverse Fourier transform to obtain the output.
>
> Each transformation in this pipeline is orthogonal. Specifically, we compute
>
> $$
> Y=\\text{FFT}\^{-1}[\\tilde{Y}], \\text{where } \\tilde{Y}\_{:,i,j}=\\tilde{W}\_{:,:,i,j}\\tilde{X}\_{:,i,j},\\
> $$
>
> and $\\tilde{W}\_{:,:,i,j}=I - 2\\tilde{V}\_{:,:,i,j}(\\tilde{V}\_{:,:,i,j}^* \\tilde{V}\_{:,:,i,j})^{-1}\\tilde{V}\_{:,:,i,j}^*$ for each pixel index $i$ and $j$.
> This process demonstrates that $X$ undergoes a series of orthogonal operations, thereby preserving the norm of $X$ through BRO convolution. We have rephrased the statements from lines 200 to 215 and revised Algorithm 1 to enhance clarity and precision.
>
> To further verify that BRO convolution is orthogonal, follwoing Cayley, we conduct some empirical test on the value of $\lVert \text{BRO}(x) \rVert / \lVert x \rVert$ for a trained BRO convolution layer. The following table show the statistics of 1000 forward of random generated input $x$.
>
> | Statistics | Values   |
> | ---------- | -------- |
> | Mean       | 0.997850 |
> | Std        | 0.000088 |
>
> **Q: Why do we need FFT and the convolution theorem?**
>
> BRO convolution, $W_{\text{Conv}} = I_{\text{Conv}} - 2V \circledast (V^{\top} \circledast V)^{-1} \circledast V^{\top}$, involves an inverse convolution term $(V^{\top} \circledast V)^{-1}$, which is difficult to compute it in the original spatial domain. Using the Fourier transform and the convolution theorem allows us to compute this efficiently.
>
> **Q: Question about $c'_\text{in}$ in Algorithm 1**
>
> We would like to firstly clarify that $c_\text{in}'$ represents the low-rank dimension of the unconstrained parameter $V \in \mathbb{R}^{c_\text{out} \times c_\text{in}' \times k \times k}$, which is not the same as those (with number of dimension $c_\text{in}$ and $c_\text{out}$) used to perform convolution on the input.
>
> Since orthogonalization is applied to the channel dimensions, we intentionally initialize the input channels of the parameter kernel to be fewer than the actual input channels to align with the design of the BRO dense layer. That's why we argue $c_\text{in}' \leq c_\text{in}$. This is the key feature that makes BRO convolution to be low-rank parameterized. Notably, after parameterization, the channel dimensions of $W$ reverts to the actual number of channels.
>
> For improved readability, we have updated the notations by redefining $c_\text{in}$ and $c_\text{out}$ as the unified notation $c$ and the low-rank dimension as $n$.
>
>
> **Q: Question about the semi-orthogonal case**
>
> For the BRO dense layer, given an unconstrained matrix $V\in\mathbb{R}^{m \times n}$, the resulting parameterized matrix $W\in\mathbb{R}^{m \times m}$ will be truncated via  $W[:d_{\text{out}},:d_{\text{in}}]\in\mathbb{R}^{d_{\text{out}} \times d_{\text{in}}}$, where $d_{\text{in}}$ and $d_{\text{out}}$ represent the input and output dimension, and $m=\max(d_{\text{out}}, d_{\text{in}})$.
>
> For the BRO convolutional case, since orthogonalization is performed along the channel dimensions, a similar truncation approach is applied to the channel dimensions.
>
> **Q: Question about Equation 14 in the proof of Proposition 2**
>
> The condition for the equation to hold is satisfied.
>
> Note that for invertible matrices with $A, B \in \mathbb{C}^{m \times m}$, $(ABA^*)^{-1} = (A^*)^{-1} B^{-1} A^{-1}$.
>
> Thus, we have
> $$
>     (\tilde{V}^* \tilde{V})^{-1}
>     = J^* (J^*)^{-1} (\tilde{V}^* \tilde{V})^{-1} J^{-1} J
>     = J^* (J \tilde{V}^* \tilde{V} J^*)^{-1} J
>     \neq J (J \tilde{V}^* \tilde{V} J^*)^{-1} J^*
> $$

---

> ### Author Response · Authors · 2024-11-18
>
> **Q: Regarding LA and CR**
>
> While training Lipschitz models, we observed that the logit margin would saturate, further improvement becomes challenging. To explain this phenomenon, we derived Theorem 1 and Proposition 3. These results show that Lipschitz models inherently struggle to minimize margin risk. To address this, we propose LA to dynamically controls the logit margin of individual data points during training. Data points with large margins are gradually annealed. While we acknowledge that this design is heuristic, it effectively ensures that all data points achieve a fair and appropriate logit margin. We will fully state the CR loss and compare it with LA.
>
>
> **Q: Extending Table 2 Experiments**
>
> Thank you for your question. We have conducted new experiment for CIFAR-10+EDM under this setting, and the results are as follows:
>
> | Model           | Clean    | $\varepsilon=36/255$ | $\varepsilon=72/255$ | $\varepsilon=108/255$ |
> | --------------- | -------- | -------------------- | -------------------- | --------------------- |
> | LiResNet        | 87.0     | 78.1                 | 66.1                 | 53.1                  |
> | LiResNet+BRO    | **87.4** | **78.4**             | 66.8                 | 53.4                  |
> | LiResNet+LA     | 86.7     | 78.1                 | 67.0                 | 54.2                  |
> | LiResNet+LA+BRO | 87.2     | 78.3                 | **67.4**             | **54.5**              |
>
> We observed that using the BRO backbone enhances performance. However, similar to the baseline LiResNet, BRONet with LA does not show improvements in clean or $\varepsilon=36/255$ certified accuracy in the CIFAR-10+EDM scenario. This may be because the learning dynamics with an abundance of diffusion-synthetic images (80 times more than real images, with a 1:3 real-to-synthetic ratio in each batch) are less impacted by the challenges discussed in the LA section. Moreover, the LA hyperparameters were initially selected on LipConvNet without incorporating diffusion-synthetic images, which could result in suboptimal alignment with the current setting.

---

> > ### Comment · Reviewer_1mnT · 2024-11-20
> >
> > Dear authors
> >
> > I am sorry, but after your response, even more concerns appear and the previous ones have not been addressed.
> >
> > ## New points
> >
> > - **You define the convolutional kernel as $W_{\text{Conv}} = I_{\text{Conv}} - 2V \circledast (V^{\top} \circledast V)^{-1} \circledast V^{\top}$**
> >
> > There is again no proof that this kernel is orthogonal or that $\text{FFT}(W_{\text{Conv}}) = I - 2\tilde{V}(\tilde{V}^{\*}\tilde{V})^{-1}\tilde{V}^{\*}$. Also $\text{FFT}(I_{\text{Conv}}) \neq I$. Please provide explicit proofs in the next version of your paper.
> >
> > - **Your first claim in the introduction is that "We propose a novel BRO method to construct orthogonal layers ... stable during training by eliminating the need for iterative approximation algorithms"**
> >
> > If your method is exactly orthogonal (no need of approximations), how come $||\text{BRO}(X)||/||X||\neq 1$ in your newly added results?
> >
> > ## Old points
> >
> > - **On the semi-orthogonal case**
> >
> > Your explanations are very vague: "a similar truncation approach is applied to the channel dimensions". Again, how does truncation affect orthogonality? How is it handled in Algorithm 1?
> >
> > - **What are the consequences of Proposition 2?**
> >
> > It is still unclear what are the consequences of this proposition regarding orthogonality. If this is something needed to apply a theorem/lemma in previous work, please explicitly mention it.
> >
> > - **Regarding LA and CR**
> >
> > You mention "We will fully state the CR loss and compare it with LA". It has not been added in the revised manuscript.
> >
> > I believe authors should invest their efforts in explicitly proving orthogonality of their BRO convolution up to *every detail in their models*. The whole point of the certified robustness field is to have theoretical guarantees of the robustness of our models. BRO convolution increases the speed and the performance, but until orthogonality is explicitly clear, every empirical result is invalid. After checking that the previous concerns were not addressed and new concerns arised, I am forced to decrease my score.

---

> ### Author Response · Authors · 2024-11-22
>
> Dear Reviewer,
>
> Thank you for your valuable feedback. We have revised the paper accordingly, making updates to Section 4.1, Proposition 2, and Appendix A.2. Additionally, we have added Appendices A.3 and A.4 to clarify specific techniques employed in the BRO convolution.
>
> **Q: Regarding Proposition 2**
>
> The original Proposition 2 was used to prove that, despite BRO convolution being performed in the Fourier domain (involving complex numbers), the output remains real.
>
> For enhanced clarity, we have moved the original Proposition 2 to Appendix A.2, where it is now stated as Lemma 3, and introduced a new Proposition 2 to demonstrate that the BRO convolution is orthogonal while ensuring a real output.
>
> **Q: Explicit Proof for Orthogonality**
>
> Explicit proof for Proposition 2 is added in Appendix A.2 to demonstrate that the BRO convolution is orthogonal. The proof is based on the idea that BRO convolution applies a sequence of orthogonal operations to the vectorized input, thereby ensuring its orthogonality.
>
> Lastly, for the circular convolution identity kernel $I_{\text{Conv}}$, as the matrix and vector multiplications are performed along the channel dimensions instead of pixel dimensions, the matrix $\text{FFT}(I_{\text{Conv}})_{:,:,i,j}$ is indeed equivalent to $I$ in the Fourier domain for every pixel index $i$ and $j$.
>
> **Q: Empirical Error Gap on Orthogonality**
>
> Our analysis reveals that the slight norm drop is caused by numerical errors and the zero-padding trick used in lines 2 and 3 of Algorithm 1, followed by the removal of padded pixels in line 8. Following LOT [1], we apply zero-padding on images $X$, creating $X_{\text{pad}}$, before performing the 2D $\text{FFT}$. After applying $\text{FFT}^{-1}$, we obtain $Y\_{\\text{pad}}$, from which the padded pixels are removed to restore the original dimensions of $X$. This approach leverages zero-padding to avoid circular convolution across edges, which empirically improves performance [1]. However, norm preservation only holds for $||X||=||X\_{\text{pad}}||=||Y\_{\text{pad}}||$. Consequently, removing pixels from $Y_{\text{pad}}$ causes the slight norm drop. Importantly, it does not affect the validity of the certified results, as neither zero-padding nor the removal of padded parts expands the norm or violates the 1-Lipschitz bound.
>
> We have incorporated this clarification into lines 236 to 240 and a detailed discusssion in Appendix A.4 of the revised paper. Additionally, for reference, we provide the empirical test for $\lVert \text{BRO}(x) \rVert / \lVert x \rVert$ when zero-padding is omitted:
>
> | Statistics | Values             |
> | ---------- | ------------------ |
> | Mean       | 0.9999998812079429 |
> | Std        | 0.0000000339773139 |
> | Max        | 1.0000000000000000 |
> | Min        | 0.9999997615814209 |
>
> **Q: On the Semi-orthogonal Case for BRO Convolution**
>
> As previously explained, truncation occurs along the channel dimensions: For each pixel index $i$ and $j$, where $c = \max(c_{\text{out}}, c_{\text{in}})$, we parameterize $\tilde{V}\_{:,:,i,j} \in \mathbb{C}^{c \times n}$ as $\tilde{W}\_{:,:,i,j} \in \mathbb{C}^{c \times c},$ then truncate it to $\tilde{W}\_{:c_{\text{out}},:c_{\text{in}},i,j} \in \mathbb{C}^{c_{\text{out}} \times c_{\text{in}}}$.
>
> We have now explicitly include this in line 244 to 247 of the revised paper, and discuss how semi-orthogonal affect orthogonality in Appendix A.3.
>
> **Q: Regarding CR and LA**
>
> We sincerely apologize for the oversight. In the initial version of the paper, we introduced the CR formula, $\mathcal{L}_{\mathrm{CE}} - \gamma \mathrm{ReLU}(\mathcal{M}_f(x))$, at the beginning of Section 6 (Line 312). To improve clarity, we have now reintroduced $\mathcal{M}_f(x)$ here, which was originally defined at the end of Section 2.1. Additionally, to enhance readability and ease of reference, we have restated the CR term before discussing it in Appendix C.2.
>
> Thank you for your valuable comments. We have made every effort to revise the paper for enhanced clarity. Please let us know if there are any additional concerns or issues that require our attention.
>
> [1] Xu et al. "LOT: Layer-wise Orthogonal Training on Improving l2 Certified Robustness." Advances in Neural Information Processing Systems (NeurIPS), 2022.

---

> > ### Comment · Reviewer_1mnT · 2024-11-25
> >
> > Dear authors,
> >
> > Thanks for your efforts in making your orthogonality proof more formal, specially with the short time given.
> >
> > However, I think the proof is far from clear. Let me explain.
> >
> > - **Regarding $W_{\text{Conv}} = I_{\text{Conv}} - 2V \circledast (V^{\top} \circledast V)^{-1} \circledast V^{\top}$**
> >
> > Why is convolving with this kernel equivalent to algorithm 1? Please **explicitly** compute $\text{FFT}(W_{\text{Conv}})$. I assume that convolution with $I_{\text{Conv}}$ results in the identity mapping. Why is that $\text{FFT}(I_{\text{Conv}})_{:,:,i,j} = I$? What is the point of presenting this kernel?
> >
> > I do not understand why is this kernel is even presented. I have repeatedly asked about this convolutional kernel and its relationship with Algorithm 1 in my previous responses and the responses I got where not satisfactory.
> >
> > - **Regarding Appendix A.2**
> >
> > I appreciate the inclusion of this section. Now the reader can understand where authors want to arrive to (Equation 17). Nevertheless, the proof is far from rigorous. Lines 864-869 are very unclear:
> >
> > - **Why is $\text{BRO}(\mathcal{D}) = \text{diag}(\text{BRO}(\mathcal{D} _1), ..., \text{BRO}(\mathcal{D} _{s^2}))$**
> >
> > How do you even build a diagonal matrix from a list of matrices?
> >
> > - **Regarding Appendix A.3 and A.4**
> >
> > I appreciate the inclusion of these appendices, now it is clear why the BRO layer is not exactly norm-preserving in some cases but is still 1-Lipschitz.
> >
> > Overall, I appreciate the efforts of the authors. I will increase back to 3 because I think it is not fair to give a 1. I believe the paper still needs a lot of work to make the claims very clear and improve the presentation. While I can now "believe" the BRO layer is 1-Lipschitz, the result is far from being clearly presented in the paper and not up to the quality standards of the conference. I still believe the paper is interesting and can be a good work if revised. But, currently, I cannot propose the paper for acceptance.

---

> > > ### Author Response · Authors · 2024-11-26
> > >
> > > **Q: Regarding $W_\text{Conv}$**
> > >
> > > Thank you for your valuable comments.
> > >
> > > Broadly speaking, the 2D convolution theorem [1] bridges the spatial domain and the Fourier domain, establishing an *equivalence* between spatial convolution and point-wise multiplication in the Fourier domain. Since constructing BRO solely in the spatial domain is challenging, as mentioned in the manuscript, our method instead constructs BRO using operations in the Fourier domain. Thus, presenting the spatial domain kernel aims to facilitate the reader's understanding of our goal.
> > >
> > > To clarify,
> > > $$
> > > \text{FFT}(W\_\text{Conv})\_{:,:,i,j} = \tilde{W}\_{:, :, i, j} = (I - 2 \tilde{V}\_{:,:,i,j} (\tilde{V}^*\_{:,:,i,j}\tilde{V}\_{:,:,i,j})^{-1} \tilde{V}^{*}\_{:,:,i,j}),
> > > $$
> > > for each pixel indices $i$ and $j$.
> > >
> > > According to the convolution theorem [1], in the Fourier domain the $c^{th}$ output channel is the sum of the element-wise products of the $c_\text{in}$ input and weight channels. That is,
> > >
> > > $$
> > > \tilde{Y}\_{c,:,:} = \sum_{k=1}^{c_{in}} \tilde{W}\_{c, k, :,:} \odot \tilde{X}_{k,:,:} ~,
> > > $$
> > > where we use the same notations in Appendix A.2. Equivalently, the $(i,j)^{th}$ pixel of the $c^{th}$ output channel is the dot product of the $(i,j)^{th}$ pixel of the $c^{th}$ weight with the $(i,j)^{th}$ input pixel:
> > >
> > > $$
> > > \tilde{Y}\_{c, i, j} = \tilde{W}\_{c, :, i, j} \cdot \tilde{X}\_{:, i, j}.
> > > $$
> > >
> > > From this, we can see that the whole $(i,j)^{th}$ Fourier-domain output pixel is the matrix-vector product:
> > >
> > > \begin{align}
> > > \text{FFT}(W\_{\text{Conv}} \circledast X)\_{:, i, j} &= \tilde{W}\_{:, :, i, j} \tilde{X}\_{:, i, j} \\\\
> > > &= (I - 2 \tilde{V}\_{:,:,i,j} (\tilde{V}^*\_{:,:,i,j}\tilde{V}\_{:,:,i,j})^{-1} \tilde{V}^{*}\_{:,:,i,j}) \tilde{X}\_{:, i, j}.
> > > \end{align}
> > > That is, convolving with $W_{\text{Conv}}$ is equivalent to line 6 in Algorithm 1.
> > >
> > >
> > >
> > > Next, we explain $I_{\text{Conv}}$. The identity kernel $I_{\text{Conv}} \in \mathbb{R}^{c \times c \times s \times s}$ for circular convolution is:
> > > $$
> > > I_{\text{Conv}}[k, l, h, w] =
> > > \begin{cases}
> > > 1, & \text{if } h = 0, w = 0 \text{ and } k = l, \\\\
> > > 0, & \text{otherwise}.
> > > \end{cases}
> > > $$
> > > When the 2D FFT is applied to this matrix, it becomes:
> > > $$
> > > \tilde{I}\_{\text{Conv}} [k, l, :, :] = \text{FFT}(I\_{\text{Conv}}[k, l, :, :]),
> > > $$
> > > for each channel indices $k$ and $l$.
> > >
> > > Let $\omega = e^{-j \frac{2\pi}{N}}$, when $k = l$, the value of $\text{FFT}(I_{\text{Conv}}[k, l, :, :])$ can be computed as follows,
> > > $$
> > > \frac{1}{N}
> > > \begin{bmatrix}
> > > 1 & 1 & \cdots & 1 \\\\
> > > 1 & \omega & \cdots & \omega^{N-1} \\\\
> > > 1 & \omega^2 & \cdots & \omega^{2(N-1)} \\\\
> > > \vdots & \vdots & \ddots & \vdots \\\\
> > > 1 & \omega^{N-1} & \cdots & \omega^{(N-1)^2}
> > > \end{bmatrix}
> > > \cdot
> > > \begin{bmatrix}
> > > 1 & 0 & \cdots & 0 \\\\
> > > 0 & 0 & \cdots & 0 \\\\
> > > \vdots & \vdots & \ddots & \vdots \\\\
> > > 0 & 0 & \cdots & 0 \\\\
> > > \end{bmatrix}
> > > \cdot
> > > \begin{bmatrix}
> > > 1 & 1 & \cdots & 1 \\\\
> > > 1 & \omega & \cdots & \omega^{N-1} \\\\
> > > 1 & \omega^2 & \cdots & \omega^{2(N-1)} \\\\
> > > \vdots & \vdots & \ddots & \vdots \\\\
> > > 1 & \omega^{N-1} & \cdots & \omega^{(N-1)^2}
> > > \end{bmatrix}
> > > = \begin{bmatrix}
> > > 1 & 1 & \cdots & 1 \\\\
> > > 1 & 1 & \cdots & 1 \\\\
> > > \vdots & \vdots & \ddots & \vdots \\\\
> > > 1 & 1 & \cdots & 1 \\\\
> > > \end{bmatrix}.
> > > $$
> > >
> > > For the cases that $k \neq l$, since the values in $I\_{\text{Conv}} [k, l, :, :]$ are all zeros, the resulting matrices $\tilde{I}\_{\text{Conv}} [k, l, :, :]$ are zero matrices.
> > >
> > > Thus,
> > > $$
> > > \tilde{I}_{\text{Conv}} [k, l, i, j] =
> > > \begin{cases}
> > > 1, & \text{if } k=l, \\\\
> > > 0, & \text{otherwise}.
> > > \end{cases}
> > > $$
> > > Therefore, for each $\tilde{I}\_{\text{Conv}} [:, :, i, j]$, it is exactly an identity matrix $I$.
> > >
> > >
> > > In conclusion, although $W\_\text{Conv}$ and $I\_\text{Conv}$ are not required to implement BRO convolution in Fourier domain, presenting the spatial domain kernels $W\_\text{Conv}$ and $I\_\text{Conv}$ helps facilitate the reader's understanding of the equivalence between spatial and Fourier domain representations.
> > >
> > >
> > > **Q: Regarding Appendix A.2**
> > >
> > > We apologize for the confusion caused by the notation. The operator $\text{diag}$ is intended to represent the **direct sum** of these matrices. We have now revised the equation as follows:
> > > $$\text{BRO}(\mathcal{D}) = \text{BRO}(\mathcal{D}\_1) \oplus \cdots \oplus \text{BRO}(\mathcal{D}\_{s^2}).
> > > $$
> > >
> > >
> > > [1] Anil K Jain. Fundamentals of digital image processing. Prentice-Hall, Inc., 1989.

---

> > > > ### Comment · Reviewer_1mnT · 2024-12-02
> > > > **Thanks for your engagement**
> > > >
> > > > Dear Authors,
> > > >
> > > > Thanks for your clarifications and engagement during this rebuttal. Please find my last comments here:
> > > >
> > > > In the next version of the paper, please work on the proof of the following expression:
> > > >
> > > > $\text{FFT}(W _{\text{Conv}}) _{:,:,i,j} = \tilde{W} _{:, :, i, j} = (I - 2 \tilde{V} _{:,:,i,j}(\tilde{V}^{\*} _{:,:,i,j}\tilde{V} _{:,:,i,j})^{-1}\tilde{V}^{\*} _{:,:,i,j}).$
> > > >
> > > > At the moment, I believe this is wrong. If so, please work on coming up with the inverse FFT of the BRO convolution. This way you can get the appropriate kernel you should use in the non-fourier domain. Also, please work on providing a formal proof of orthogonality. Define the dimensions of each block $\mathcal{D}_i$, define the direct sum $\oplus$ and make everything clear to the reader. I will keep my score.
> > > >
> > > > Regards,
> > > >
> > > > Reviewer 1mnT

---

> > > > > ### Author Response · Authors · 2024-12-04
> > > > >
> > > > > Dear Reviewer,
> > > > >
> > > > > Thank you for your effort during this rebuttal. We would like to emphasize that most of your concerns are directly addressed by the well-established correctness of the FFT+convolution theorem technique [1][2].
> > > > >
> > > > > To clarify, the proposed BRO convolution method parameterizes an orthogonal weight matrix in the Fourier domain. After obtaining $\tilde{W}$ and $\tilde{X}$ in the Fourier domain, we apply the the established FFT+convolution technique to perform the orthogonal operations efficiently and only invert $\tilde{Y}$ to get the output results.
> > > > >
> > > > > Importantly, despite being feasible, there is no need to invert $\tilde{W}$ to compute $W_\text{conv}$ in order to obtain the output $Y$. The latter serves solely as an equivalent definition in the spatial domain to facilitate understanding. All derivations remain correct based on the established FFT+convolution theorem technique.
> > > > >
> > > > > Sincerely,
> > > > >
> > > > > Authors
> > > > >
> > > > > [1]: Trockman et al. "Orthogonalizing Convolutional Layers with the Cayley Transform." International Conference on Learning Representations(ICLR), 2020.
> > > > >
> > > > > [2]: Xu et al. "LOT: Layer-wise Orthogonal Training on Improving l2 Certified Robustness." Advances in Neural Information Processing Systems (NeurIPS), 2022.

---

### Official Review · Reviewer_qcmE · 2024-10-29

**Soundness:** 3
**Presentation:** 3
**Contribution:** 3
**Rating:** 6
**Confidence:** 3

**Summary:**

This paper introduces a novel approach called Block Reflector Orthogonal (BRO) layer for constructing Lipschitz neural networks with certified robustness guarantees against adversarial attacks. The key innovation is a new parameterization scheme that creates orthogonal layers without requiring iterative approximation algorithms, making it both computationally efficient and numerically stable compared to existing methods like SOC and LOT. The authors use BRO to develop BRONet, which achieves state-of-the-art certified robustness on CIFAR-10, CIFAR-100, and Tiny-ImageNet datasets. Additionally, they provide theoretical analysis showing that Lipschitz networks have inherent limitations in margin maximization due to limited model complexity, leading them to propose a new Logit Annealing (LA) loss function that employs an annealing mechanism to help models learn appropriate margins for most data points rather than overfitting to maximize margins for specific examples. Through extensive experiments, they demonstrate that their combined approach (BRO layer + LA loss) outperforms existing methods while requiring fewer parameters and computational resources.

**Strengths:**

- Strong theoretical foundations with clear mathematical proofs for their proposed methods
- Significant computational efficiency gains compared to existing approaches (SOC, LOT)
- Comprehensive experiments across multiple datasets and architectures
- Novel loss function with clear theoretical motivation and empirical benefits
- State-of-the-art certified robustness with fewer parameters
- Thorough ablation studies demonstrating impact of each component

**Weaknesses:**

- BRO layer is not a universal orthogonal parameterization (acknowledged by authors)
- Limited experiments on larger datasets beyond Tiny-ImageNet
- No comparison with empirical defenses or other certified robustness approaches beyond Lipschitz methods
- Lack of investigation into potential failure cases or limitations of the LA loss
- Some hyper-parameters (rank selection, LA loss parameters) require manual tuning

**Questions:**

- Have you tested BRO's performance on larger datasets like ImageNet?
- How sensitive is the method to the choice of rank in BRO and how should practitioners select it?
- How does the LA loss perform compared to other margin-based losses beyond CE+CR?
- What are the main failure cases or limitations of your approach?
- Have you considered comparing against other certified defense methods beyond Lipschitz approaches?

---

> ### Author Response · Authors · 2024-11-18
> **Response for Reviewer qcmE**
>
> **Q: Have BRO be tested on larger datasets like ImageNet?**
>
> Training a model on ImageNet is beyond the scope of our current computational resources. For reference, the ImageNet experiments for LiResNet (Hu et al., 2023&2024) reportedly required 8 A100 GPUs, while a single training run would take over two weeks on our current GPU devices. We believe our extensive experiments on CIFAR-10, CIFAR-100, TinyImageNet, and CIFAR-10/100+EDM sufficiently demonstrate the effectiveness of our proposed method.
>
>
> **Q: How sensitive is the method to the choice of rank in BRO and how should practitioners select it?**
>
> In summary, for practical applications, we recommend starting with half rank $n = m / 2$. This recommendation stems from the fact that the variable $n$ controls the ratio of $+1$ to $-1$ eigenvalues in the orthogonal matrix $W$, as illustrated in Proposition 1. Intuitively, a predominance of either $+1$ or $-1$ eigenvalues could negatively impact the diversity of $W$.  Additionally, we have provided an empirical demonstration of how the selection of rank affects performance in Table 9.
>
>
> **Q: Comparison between LA and other margin-based losses.**
>
> In the context of training Lipschitz neural networks, CE+CR emerges as the most effective loss function for enhancing margins, as supported by several existing studies[1][2]. During this work's development, we also explored other margin-based loss functions, such as OVR[3] and MMA[4], integrating them into the training procedure. However, their performance consistently fell short compared to CE+CR. Consequently, our primary comparison focuses on the performance of LA against CE+CR.

---

> ### Author Response · Authors · 2024-11-18
>
> **Q: The limitation of our approach.**
>
> The main limitation of the BRO layer design is that it can only tightly certify against $\ell_2$-norm attacks. Although, with some relaxation, the certification could potentially generalize to other norms, the resulting certification may lack tightness and become less effective. For the LA loss, the need for hyperparameter tuning introduces further challenges. We will explicitly acknowledge these limitations in the paper.
>
>
> **Q: Comparison with certified defense methods beyond Lipschitz approaches.**
>
> To the best of our knowledge, we have listed out all the competitive deterministic $\ell_2$ certified defense methods.
>
> There are other certified defense methods that aim to provide robust guarantees with respect to different norms. For instance, Interval Bound Propagation (IBP) was originally designed for $\ell_\infty$-norm defenses. Though IBP can also perform relaxation and provide $\ell_2$-norm certified robustness, the $\ell_2$-norm guarantees it offers are generally loose[5]. Therefore, we did not include these comparisons in our paper.
>
> However, we can compare the empirical robustness of those different methods. For reference, we tested our model on CIFAR-10 against the standard AutoAttack[6] with perturbation budgets of $\ell_{\infty}=2/255$ and $8/255$. Note that our method is designed for certifying against the $\ell_{2}$-norm based attacks, whereas the reported IBP-based baselines are designed for the $\ell_{\infty}$ attacks. The baseline results are reported from the literature[7][8].
>
>
> | Method | Clean| AutoAttack $\ell_{\infty}=2/255$ | AutoAttack $\ell_{\infty}=8/255$ | Certification  |
> |-|-|-|-| - |
> | STAPS $\ell_{\infty}=2/255$ | 79.75| 65.91   | N/A | 62.72 ($\ell_{\infty}=2/255$)|
> | SABR $\ell_{\infty}=2/255$  | 79.52| 65.76   | N/A | 62.57 ($\ell_{\infty}=2/255$)|
> | IBP $\ell_{\infty}=8/255$   | 48.94| N/A | 35.43| 35.30 ($\ell_{\infty}=8/255$)|
> | TAPS $\ell_{\infty}=8/255$  | 49.07| N/A | 34.75| 34.57 ($\ell_{\infty}=8/255$)|
> | SABR $\ell_{\infty}=8/255$  | 52.00| N/A | **35.70**| 35.25 ($\ell_{\infty}=8/255$)|
> | BRONet-L (Ours) | **81.57** | **68.76** | 21.02| 70.59, 57.15 , 42.53 ($\ell_{2}=36/255,72/255,108/255)$ |
>
> Another approach to achieving $\ell_2$ certified robustness is randomized smoothing. However, unlike Lipschitz networks, its certification is inherently probabilistic and not directly comparable, so we did not focus on comparing with it.
>
> [1] Yu et al. "Constructing orthogonal convolutions in an explicit manner." International Conference on Learning Representations (ICLR), 2022.
>
> [2] Xu et al. "LOT: Layer-wise Orthogonal Training on Improving l2 Certified Robustness." Advances in Neural Information Processing Systems (NeurIPS), 2022.
>
> [3] Kanai et al. "One-vs-the-rest loss to focus on important samples in adversarial training." International Conference on Machine Learning (ICML), 2023.
>
> [4] Ding et al. "MMA Training: Direct Input Space Margin Maximization through Adversarial Training."International Conference on Learning Representations (ICLR), 2020.
>
> [5] Li et al. "Sok: Certified robustness for deep neural networks." 2023 IEEE symposium on security and privacy (SP), 2023.
>
> [6] Croce, et al. "Reliable evaluation of adversarial robustness with an ensemble of diverse parameter-free attacks." International Conference on Machine Learning (ICML), 2020.
>
> [7] Mao, et al. "Connecting certified and adversarial training." Advances in Neural Information Processing Systems (NeurIPS), 2023.
>
> [8] Mueller, et al."Certified training: Small boxes are all you need." International Conference on Learning Representations (ICLR), 2023.

---

> > ### Comment · Reviewer_qcmE · 2024-11-25
> >
> > Thank you for your response. I will keep my score as is.

---

> > > ### Author Response · Authors · 2024-11-25
> > >
> > > We sincerely appreciate your positive feedback and recognition. Thank you for taking the time and effort to provide such thoughtful and constructive comments.

---

### Official Review · Reviewer_2jVt · 2024-11-01

**Soundness:** 3
**Presentation:** 3
**Contribution:** 3
**Rating:** 6
**Confidence:** 2

**Summary:**

This work presents a novel BRO layer for constructing Lipschitz neural networks that leverages low-rank parameterization, avoiding iterative approximations to enhance both memory and computational efficiency. The introduction of the Logit Annealing loss function addresses the complexity limitations of Lipschitz networks, contributing to improved learning of margin properties. This work promises good advancements in robust and efficient Lipschitz neural networks design.

**Strengths:**

1. The proposed method is supported by good theoretical analysis.

2. The BRO improves both efficiency and stability compared with previous work.

3. The proposed Annealing Loss could serve as a scalable method to enhance the training for the Lipschitz neural network.

**Weaknesses:**

1. **The scalability to other large models**: The proposed BRO layer is currently designed to function within specific neural network architectures, such as the BRONet Architecture, maybe with constrained parameters and fixed configurations. This limitation raises concerns about its practical applicability to large-scale foundation models. Ensuring the Lipschitz condition for these more complex and expansive models could be challenging. Nonetheless, the impact of this work would be significantly enhanced if the authors could empirically demonstrate that BRO effectively improves the robustness of more complex and diverse neural networks.

2. According to lines 276-279, the BRO layer is not a universal approximation for orthogonal layers and the author empirically demonstrates BRO is competitive with that of LOT and SOC. A more detailed analysis would be beneficial to illustrate that the error introduced by this non-universal approximation is minimal.

3. I am curious about whether the choice of activation function affects the Lipschitz constant.

**Questions:**

See weakness.

---

> ### Author Response · Authors · 2024-11-18
> **Response for Reviewer 2jVt**
>
> **Q: The potential for large and complex models, such as foundation models.**
>
> Thank you for the insightful suggestion. Extending our method to large-scale foundation models is indeed an intriguing direction, and we consider it an important avenue for future research. Recent studies have also begun to investigate the Lipschitz continuity of more complex models [1]. Developing a Lipschitz foundation model with certified robustness could have significant implications, and we hope our research provides a solid foundation for future advancements in this area.
>
> **Q: Non-universal orthogonal approximation for BRO.**
>
> To assess the potential negative impacts arising from BRO's non-universal orthogonal approximation property, we conducted an experiment on the MNIST dataset using the simplest possible setting: a linear model. *Since the model's capacity is directly tied to the linear orthogonal matrices that BRO can represent in this setting, the experiment aims to determine whether BRO's performance deteriorates due to this inherent property.*
>
> The experiment consists one single-layer comparison (utilizing a single BRO/LOT linear layer).
>
> |     | Train Acc. | Valid Acc. | Cert. Acc. (36/255) | Cert. Acc. (72/255) | Cert. Acc. (108/255) |
> | --- | ---------- | ---------- | ------------------- | ------------------- | -------------------- |
> | BRO | 90.44%     | 90.73%     | 89.45%              | 88.34%              | 87.10%               |
> | LOT | 89.58%     | 90.19%     | 88.85%              | 87.66%              | 85.90%               |
>
> Note that LOT is a universal orthogonal approximator. While BRO exhibits weaker representation ability than LOT in a single-layer setting, it achieves comparable or even favorable accuracies. We suggest that this may be attributed to optimization bias or other factors influencing the learning dynamics, which outweight the disadvantages of the non-universal approximation orthogonal property. In conclusion, the non-universal orthogonal approximation property does not make BRO less competitive compared to other methods.
>
>
> The experiment details are provided as follows.
>
> | Hyperparameter            | Value         |
> | ------------------------- | ------------- |
> | Optimizer                 | Adam          |
> | Loss function             | Cross Entropy |
> | Batch size                | 64            |
> | Learning rate             | 0.001         |
> | #epochs                   | 10            |
> | BRO $n$-rank   ratio      | 0.5           |
>
>
> **Q: The choice of activation & Lipschitz constant**
>
> The Lipschitz constant of common element-wise activations depend on its maximum slope, which would affect the Lipschitz constant of the entire networks. For example, ReLU is 1-Lipschitz as the maximum slope is one. However, most of them are not norm-preserving, which can lead to gradient norm vanishing[2] when training Lipschitz networks. In the $\ell_2$ certified robustness literature, the MaxMin activation is a popular choice.
>
> [1] Castin et. al. "How Smooth Is Attention?." International Conference on Machine Learning (ICML), 2024.
>
> [2] Cem Anil et al. "Sorting out Lipschitz function approximation." International Conference on Machine Learning (ICML), 2019.

---

> > ### Comment · Reviewer_2jVt · 2024-11-19
> >
> > Thanks for the author's detailed response. I do not have further questions. I would like to keep my original rate marginally accepted.

---

> ### Author Response · Authors · 2024-11-19
>
> We sincerely thank you for your positive feedback and recognition. We deeply appreciate the time and effort you dedicated to providing constructive comments.

---

### Official Review · Reviewer_3TcG · 2024-11-01

**Soundness:** 2
**Presentation:** 2
**Contribution:** 2
**Rating:** 3
**Confidence:** 4

**Summary:**

This paper propose a new orthogonal parametrization, Block Reflector Orthogonal layer (BRO), that can be used in the context of Lipschitz neural networks and provide certified robustness against adversarial attacks. The authors state that the BRO method to construct orthogonal layers using low-rank parameterization is both time and memory efficient, while also being stable during training as it does need
an iterative approximation algorithms. The authors also propose a theoretical analysis and develop a novel loss function, Logit Annealing loss, to increase certified accuracy. The authors perform an extensive set of experiments to demonstrate their finds.

**Strengths:**

- The paper is well written. The motivation is clear and the contribution BRO is well stated.
- The BRO method leverage FFT for convolution in a similar fashion as the Caley approach
- The authors have performed an extensive set of experiments to demonstrate their results (comparaison and ablation study)

**Weaknesses:**

- BRO is not compared to the Cayley approach in Figure 2. Does BRO offer better memory and runtime against Cayley? can Cayley orthogonal layers appear in Figure 2?
- For the comparison on certified accuracy, there are a lot of moving parts in the experimental section and this tends to become confusing, I would suggest the authors to simplify the experiments and focus on the comparison with the state of the art, which is LiResNet (Hu et al. 2024).
- It seems that the authors did not take the results of the latest version of Hu et al. 2024 (which came out in June 2024) as there seems to be a huge gap in the reported results. Can the authors comment on this?
- Table 1 reports the first results of SLL, but there was an erratum in the latest version and the results have been revised (see Table 7 in the arxiv version of the paper).
- It seems that BRO does not perform very well at large radius, is this due to the parameterization of the loss?
- Hu et al. 2024 have shown that their approach scales to ImageNet, can the authors provide certified results for ImagNet?

**Questions:**

See Weaknesses

---

> ### Author Response · Authors · 2024-11-18
> **Response for Reviewer 3TcG**
>
> **Q: Comparison with Cayley**
>
> In Figure 2, we compare LipConvNet with the most recent methods, SOC and LOT, as Cayley has either been omitted in recent literature[1] or reported to be less competitive[2]. A more detailed discussion of Cayley follows: Although Cayley performs two fewer matrix multiplications than BRO, it does not benefit from the low-rank parameterization property. When comparing the runtime and memory consumption of Cayley and BRO in the setting of Figure 2, the results are similar. However, in the LiResNet-like architecture, which uses convolution layers with a large channel width (as seen in Table 3 experiments), BRO's low-rank parameterization provides a clear advantage. To fit within memory constraints, the number of Cayley layers must be reduced, whereas BRO operates efficiently under these conditions. In conclusion, BRO is more favorable for architectures with convolutional backbones that feature large channel widths.
>
> **Q: Simplify the experiments**
>
> Thank you for your valuable suggestion. We will work on reorganizing the experimental sections to enhance readability.
>
> **Q: The reported results of LiResNet**
>
> The observed difference with the original work is due to diffusion-generated synthetic data augmentation.
>
> For Table 1, we want to fairly compare the baselines in the literature that does not use extra diffusion data augmentation. We reproduce and report results without diffusion data augmentation, as detailed in the Table 1 caption.
>
> For Table 2 and 3, we include experiments with diffusion data augmentation (CIFAR-10/100+EDM) to verify the effectiveness of our methods in this setting. For CIFAR-10, we use the original synthetic dataset pubicly-released by Hu et al. (2024); for CIFAR-100, where the original synthetic dataset is unavailable, we rely on the dataset released from Wang et al. (2023)[3].
>
> Compared to the reported results in Hu et al. (2024), a small performance gap is observed in the CIFAR-100+EDM setting due to the unavailability of the original synthetic dataset. However, since both works follow similar procedure to generate synthetic images using EDM, the comparison results should be valid in supporting the effectiveness of our proposed approach.
>
> **Q: Table 1 update**
>
> Thank you for pointing out the erratum in the latest arXiv version of SLL, we have updated Table 1 accordingly.
>
> **Q: Is limitation of BRO at large radius due to the parameterization of the loss?**
>
> As shown in our experiments, our method achieves the best performance for both clean accuracy and certified accuracy at $\varepsilon = 36/255$ but less consistent for larger perturbations. Adjusting the loss function, such as aggressively using CR or increasing the offset in LA, could enhance performance for larger perturbations. However, this often comes at the cost of clean accuracy and performance at smaller $\varepsilon$ budgets.
>
> This tendency to fit only certain examples well to achieve large certified radii is not ideal, and we aim to mitigate this issue. Our ultimate goal is to develop a robust model that maintains its natural classification ability while simultaneously achieving favorable certified robustness as an additional benefit.
>
> Interestingly, we empirically observed that less expressive Lipschitz models tend to achieve slightly higher certfied accuracy at larger perturbation levels, as they collapse to fit specific data examples well. This improvement, though, also comes at the cost of lower clean accuracy and reduced performance at smaller $\varepsilon$ budgets.
>
>
> **Q: ImageNet Results Availability**
>
> Training any model on ImageNet is prohibitively expensive for our current resources. For reference, the ImageNet experiments of LiResNet (Hu et al. 2023&2024) are reported to be conducted with 8 A100s, and a single training run would require more than two weeks with 2 RTX A6000s. We believe that our comprehensive experiments on CIFAR-10, CIFAR-100, TinyImageNet, and CIFAR-10/100+EDM are able to demonstrate the effectiveness of our proposed method.
>
> [1] Xu et al. "LOT: Layer-wise Orthogonal Training on Improving l2 Certified Robustness." Advances in Neural Information Processing Systems (NeurIPS), 2022.
>
> [2] Singla et al. "Improved deterministic l2 robustness on CIFAR-10 and CIFAR-100." International Conference on Learning Representations (ICLR), 2022.
>
> [3] Wang et al. "Better diffusion models further improve adversarial training." International Conference on Machine Learning (ICML), 2023.

---

> > ### Comment · Reviewer_3TcG · 2024-11-26
> >
> > Thank you for your response.
> >
> > Unfortunately, I cannot recommend the paper for acceptance. I believe that a critical limitation of the work lies in the absence of results on ImageNet. While the authors argue that training on ImageNet is prohibitively expensive due to resource constraints, this explanation, though understandable, does not adequately address the concern. By omitting this result, it remains unclear whether the proposed method is capable of scaling to more complex and demanding tasks. This omission raises the possibility that the approach may not generalize well beyond smaller datasets like CIFAR-10/100.
> >
> > Since the main competing approach, _i.e., LiResNet, has been shown to scale to ImageNet, I would argue that LiResNet is still a more compelling method for certified robustness.

---

> > > ### Author Response · Authors · 2024-11-26
> > >
> > > While we understand your concern regarding the absence of ImageNet results, demonstrating high certified robustness on CIFAR-10/100 is a critical benchmark and provides significant value to the field.
> > >
> > > As datasets like CIFAR-10 may appear simplistic for standard models, achieving *high certified robustness* on CIFAR-10 is far from trivial in the context of deterministic certified robustness literature. For example, current state-of-the-art methods achieve less than 80% certified accuracy for $\varepsilon = 36/255$. This highlights that even on smaller datasets, substantial challenges remain, and CIFAR-10 continues to serve as a critical benchmark for advancing foundational capabilities in certified robustness.
> > >
> > > Insights from the empirical adversarial robustness literature[1] suggest that achieving adversarial robustness on CIFAR-10 (e.g., >90% accuracy against AutoAttack at $\varepsilon_{\ell_\infty} = 8/255$) requires resources comparable to those needed for training large language models. Extending this to certified robustness on a dataset as large as ImageNet would require a prohibitively high computational budget.
> > >
> > > While we recognize LiResNet’s scalability to ImageNet as a compelling demonstration, it is important to note that certified robustness on CIFAR-10/100 remains a widely recognized and essential benchmark. Demonstrating robust performance at this level provides a strong foundation and establishes feasibility for potential future scalability to larger datasets.
> > >
> > > In addition, we have evaluated the proposed method on the Tiny-ImageNet dataset, which shares many characteristics with ImageNet but operates on a smaller scale. Although conducting experiments on ImageNet is not feasible with our current resources, we have made every effort to explore the scalability of our approach.
> > >
> > > We hope this explanation clarifies the rationale behind our experimental scope and underscores the value of our contributions within the limits of current computational feasibility. Thank you for considering our perspective.
> > >
> > > [1] Bartoldson et al., "Adversarial Robustness Limits via Scaling-Law and Human-Alignment Studies," ICML 2024.

---

### Author Response · Authors · 2024-11-18
**General Response**

We sincerely thank all the reviewers for their valuable and constructive feedback. In response to the comments, we have made  minor updates to the manuscript. The revised content is highlighted in blue for clarity.

Please read below our detailed responses to the specific comments raised.

---

### Author Response · Authors · 2024-11-25
**Revision Summary**

We deeply appreciate the reviewers' valuable feedback and their engagement in discussions with us. We have made several updates to the manuscript for enhanced clarity.

The latest revision is summarized as follows:

- To clarify the **orthogonality of BRO convolution**, we introduced a new Proposition 2, with a rigorous proof presented in Appendix A.2.
- Section 4.1 has been revised to clearly present all the details of our method.
- Appendix A.3 has been added for a detailed discussion on the semi-orthogonal case.
- Appendix A.4 has been added for a detailed discussion on the zero-padding trick.
- Appendix E. has been added to discuss the limitations of our method.

---

### Meta-Review · Area_Chair_2PJU · 2024-12-17

**Metareview:**

The paper presents the Block Reflector Orthogonal (BRO) layer, which constructs Lipschitz neural networks with enhanced certified robustness. The BRO layer's low-rank parametrization improves computational and memory efficiency, while the experiments show that this approach outperforms existing methods on standard image datasets. This seems to be a controversial paper for reviewers, since the reviewers detected major issues in the proofs and the rigor of the method. In addition, the reviewers raised multiple concerns about the claim of improved certified robustness, since this is not tested on larger datasets as more recent methods have been. Even though I do not necessarily believe that the method should be tested on ImageNet, I do agree with the reviewers on demonstrating scalability. In addition, improving rigor and the claims of the paper is very important, and therefore I do not believe the paper is yet ready to be accepted in ICLR.

**Additional Comments On Reviewer Discussion:**

The reviewers have conducted several passes over the discussion period. The authors did improve the paper everytime, but the reviewers pointed out new mistakes were made. Therefore, the paper requires another round of reviews from scratch, since this is not ready at the moment.

---

### Decision · Program_Chairs · 2025-01-22

Reject